# A common allele increases endometrial *Wnt4* expression, with antagonistic implications for pregnancy, reproductive cancers, and endometriosis

Mihaela Pavličev [1,2,3] ✉, Caitlin E. McDonough-Goldstein[2], Andreja Moset Zupan[1], Lisa Muglia[1], Yueh-Chiang Hu [1,4], Fansheng Kong[1], Nagendra Monangi [1,4], Gülay Dagdas[2], Nina Zupančič[5], Jamie Maziarz[6], Debora Sinner[1,4], Ge Zhang[1], Günter Wagner [2,6,7,8] & Louis Muglia [1,9]

The common human SNP rs3820282 is associated with multiple phenotypes including gestational length and likelihood of endometriosis and cancer, presenting a paradigmatic pleiotropic variant. Deleterious pleiotropic mutations cause the co-occurrence of disorders either within individuals, or across population. When adverse and advantageous effects are combined, pleiotropy can maintain high population frequencies of deleterious alleles. To reveal the causal molecular mechanisms of this pleiotropic SNP, we introduced this substitution into the mouse genome by CRISPR/Cas 9. Previous work showed that rs3820282 introduces a high-affinity estrogen receptor alpha-binding site at the *Wnt4* locus. Here, we show that this mutation upregulates *Wnt4* transcription in endometrial stroma, following the preovulatory estrogen peak. Effects on uterine transcription include downregulation of epithelial proliferation and induction of progesterone-regulated pro-implantation genes. We propose that these changes increase uterine permissiveness to embryo invasion, whereas they decrease resistance to invasion by cancer and endometriotic foci in other estrogen-responsive tissues.

Genome-wide association studies (GWAS) frequently reveal associations between multiple diseases and a polymorphism at a single genomic locus, suggesting pleiotropy[1]. Disease pleiotropy may originate from either the same molecular mechanism involved in different disease processes or through broader systemic effects of the variant—such as immune or metabolic disorders—affecting different disease phenotypes. While many pleiotropic variants have congruent fitness effects in all diseases (i.e. the same allele is deleterious in all), studies also commonly find cases of antagonistic pleiotropy[2], where a variant augments the likelihood of one disorder, while protecting the carrier from another another[3–5]. Antagonism in pleiotropy is particularly interesting as it may explain the high frequency of some of the apparently deleterious alleles in a population[6], reflect a biological trade-off that can inform the disease mechanisms, and provide guidance in search of treatment strategies that should be preferred or avoided due to the associated side effects[7].

[1]Cincinnati Children's Hospital Medical Center, Cincinnati, OH, USA. [2]Department of Evolutionary Biology, University of Vienna, Vienna, Austria. [3]Complexity Science Hub, Vienna, Austria. [4]University of Cincinnati College of Medicine, Cincinnati, OH, USA. [5]University Medical Center Ljubljana, Department of Cardiovascular Surgery, Ljubljana, Slovenia. [6]Department of Ecology and Evolutionary Biology, Yale University, New Haven, CT, USA. [7]Yale Systems Biology Institute, Yale University, West Haven, USA. [8]Department of Obstetrics, Gynecology and Reproductive Sciences, Yale School of Medicine, New Haven, USA. [9]Burroughs Wellcome Fund, Research Triangle Park, NC Durham, USA. ✉e-mail: mihaela.pavlicev@univie.ac.at

Deciphering the molecular basis of statistical associations requires determining the exact genomic location of the causal polymorphism, the target gene (most variants are in the regulatory genome), the spatial and temporal context of the variant effect (i.e., tissue or cell type, stage of development or physiology), the molecular pathway mediating the effect, and the organismal processes mapping the genetic change onto the final phenotype. Information from multiple diseases can facilitate the analysis of the mechanistic basis of variant effects[8]. In this paper, we present the results of a functional genetic study of one single nucleotide polymorphism (SNP) located in the first intron of *Wnt4* (rs3820282) on human chromosome 1 (1p 36.12). The frequency of the alternate allele at rs3820282 (T, with reference allele being C) varies strongly across human populations, ranging from less than 1% in Africa to over 50% in SE Asia[9]. This SNP has been associated with variation in multiple human female reproductive traits. Thereby, the same allele is associated with deleterious as well as protective effects on diseases such as endometriosis[10], breast cancer[11], ovarian epithelial cancer[12], length of gestation[13], and leiomyoma[14]. Specifically, the alternate allele has been associated with higher likelihood of endometriosis, fibroids (leiomyoma), and ovarian epithelial cancer, yet it is also associated with longer gestation and is potentially protective against preterm birth. Rather than focusing on the etiology of any one disease, we focus here on the immediate molecular and cellular mechanisms through which the locus exerts its function, likely mediating different disease contexts.

The disease associations of this SNP are consistently related to female reproductive biology, a context in which regulation by estrogen is central. A well-powered GWAS, aiming at understanding the genetic basis of preterm birth, detected association at this locus. Importantly, authors have demonstrated in vitro using electrophoretic mobility shift assay (EMSA), that the alternate allele at this locus introduces a potent binding site for estrogen receptor alpha (ESR1)[13], in accordance with computational prediction. Moreover, this binding site overlaps with open chromatin in immortalized human endometrial stromal cell line, HESC[13]. The information on the downstream consequences of this variant are less robust, although this question has been experimentally addressed previously (see discussion).

Estrogen responsiveness has been directly implicated in most diseases associated with the genomic region of the rs3820282 allele. Endometriosis has been associated with the incomplete transition of the endometrium from the estrogen-dominated proliferative phase to the progesterone-dominated secretory phase of the estrous cycle, also referred to as progesterone resistance[15], due to dysregulation in the early secretory phase of many genes mediating progesterone effects. Fine-mapping studies of the region to identify the causal SNP in the context of endometriosis have confirmed rs3820282 as a prioritized candidate[16,17]. Estrogen also enhances the growth of uterine leiomyoma (fibroids), a common benign tumor of the myometrial smooth muscle[18]. A GWAS in a large Icelandic cohort found associations of uterine fibroids with alleles at two sets of loci: loci shared with a wide range of cancers, and loci shared with various reproductive abnormalities, specifically implicating involvement of estrogen signaling, among them rs3820282[19]. Increased estrogen and androgen, as well as hyperactivity of stroma have further been proposed to contribute to the pathogenicity of ovarian cancer[20] and recent association studies linked both ovarian as well as breast cancers to the region of the focal locus[19].

This genomic region of interest is conserved across placental mammals (Fig. 1) due to the key roles of Wnts in fundamental vertebrate development and physiology, and specifically of *Wnt4* in development and physiology of the female reproductive tissues[21]. The mammalian reference genomes (including C57B/6 J mouse line, used in this study as wild type, WT) carry what in the human population is the major allele. Many aspects of female reproductive biology are also conserved between mice and humans. Therefore, we expect that the molecular effect of this variant in mice will be informative for studying its effect in humans.

To determine the molecular mechanisms affected by the SNP rs3820282, we generated CRISPR/Cas9-modified transgenic (TG) mouse line, homozygous for what is the human alternate allele, and compared it to the homozygous WT mouse line of the same background. Genome-editing circumvents the effect of linkage between the neighboring SNPs, due to which causal variants are hard to distinguish from the linked ones in the human population. This method aids in the feasibility of experimentation and allows us to avoid the heterogeneity of the genetic background between individuals, which can hinder the attribution of effects to any one polymorphism. The specific effects of a polymorphism in different contexts will need more attention in the future. Using this transgenic mouse line, we reveal that the variant enhances the periodic estrogen-induced uterine preparation for implantation. We propose that this effect, when appearing in other estrogen-responsive tissues, may similarly enhance the permissibility of tissues to metastasis and endometriotic foci.

## Results

### Uterine Wnt4 transcription in proestrus and estrus is upregulated

We used gene editing to replace the single nucleotide mouse wild type allele with the human alternate allele at the location in the mouse genome corresponding to rs3820282 in humans, using CRISPR/Cas9 (for design see Methods and Supplementary Fig. S1). The position rs3820282 and its flanking sequences are 98% conserved between the human and mouse reference genomes (Fig. 1). Live born pups were genotyped by PCR and then further confirmed by Sanger sequencing. Two out of three lines in which the overall region had been modified, had the specific one-nucleotide substitution we aimed for (referred to as KI-1 and KI-2).

Given the previous finding that rs3820282 SNP is enhancing ESR1 binding from weak to strong[19], we investigated which genes may be regulated by this novel binding site. We focused on gonadal and somatic reproductive tissues, as the previously reported associated reproductive phenotypes involved the ovary and uterus. We first investigated how the SNP affected expression of *Wnt4*, in the two independently derived knock-in lines in the ovary and uterus across multiple stages of the ovarian cycle. Specifically, we focused on proestrus and estrus, as these reflect the stages during and immediately following the estrogen peak. In the ovary, no differences in *Wnt4* expression were observed in either proestrus or estrus (Supplementary Fig. S2; proestrus: $P = 0.57$; estrus: $P = 0.79$; Supplementary Data File 1). In the uterus, the two genotypes differed in the uterine expression of *Wnt4* in proestrus with a 1.48 (KI-1) and 3.03 (KI-2) log2 fold increase, and in estrus, with a 1.61 (KI-1) and 3.27 (KI-2) log2 fold increase in the TG *Wnt4* expression compared to the wild type (Fig. 2; proestrus: $P = 0.003$ KI-1, $P = 0.0012$ KI-2; estrus: $P = 0.003$ KI-1, $P = 0.008$ KI-2; Supplementary Data File 1). There was no effect of the SNP on *Cdc42* expression, another gene adjacent to the polymorphism (Fig. 2; proestrus: $P = 0.34$ KI-1, $P = 0.63$ KI-2; estrus: $P = 1.0$ KI-1, $P = 0.80$ KI-2; Supplementary Data File 1). The consistent phenotype of increased *Wnt4* expression in both the replicate lines demonstrates the reproducible effects of the genome-edited SNP. As the KI-1 line bred more robustly than the KI-2 line, it was subsequently chosen for the full range of analyses and will be referred to by TG. To understand the systemic effects accompanying *Wnt4* increase, we analyzed the uterine transcriptome, focusing on proestrus (PE) because its final phase ends with the peak in E2 so we can directly examine the effect of the peak. Even though this time point precedes pregnancy, changes during the ovarian cycle can affect pregnancy success, as important transformations in the mouse uterus in preparation for implantation start prior to mating.

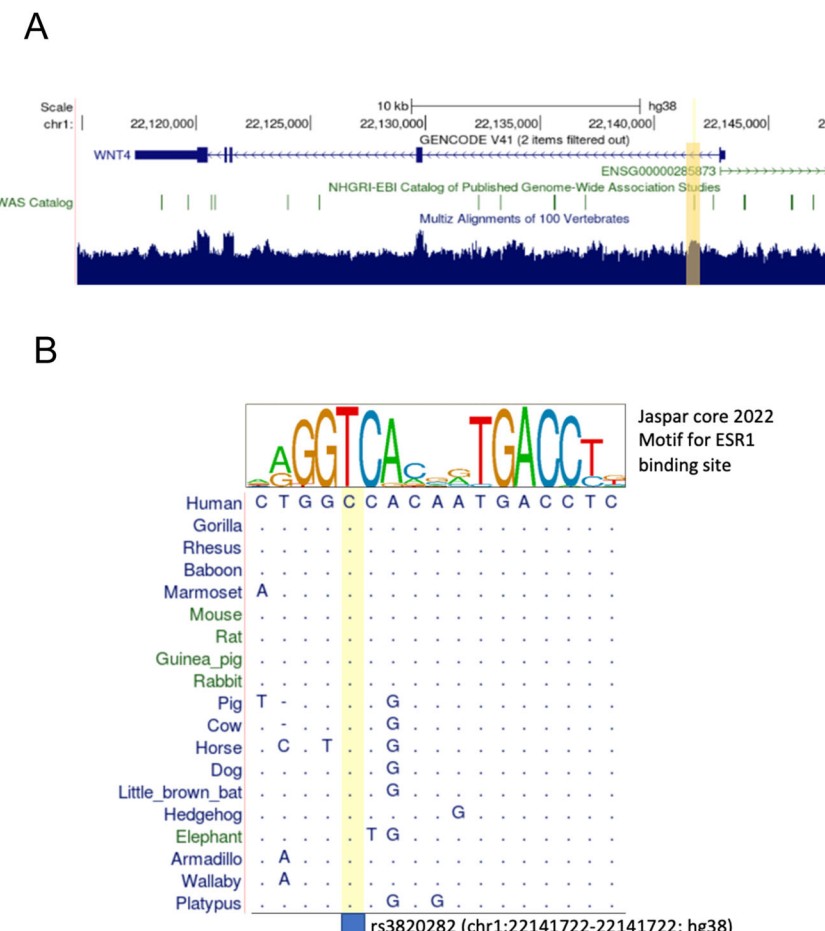

**Fig. 1 | Alignment of the noncoding region around rs3820282 across mammalian genomes (UCSC genome browser). A** overall conservation of the *Wnt4* region. **B** The binding site motif for ESR1 on top summarizes the sequence variation across the functional binding sites. The position of rs3820282 is shown in yellow in **A**, **B**, and changes the human nucleotide from C to T.

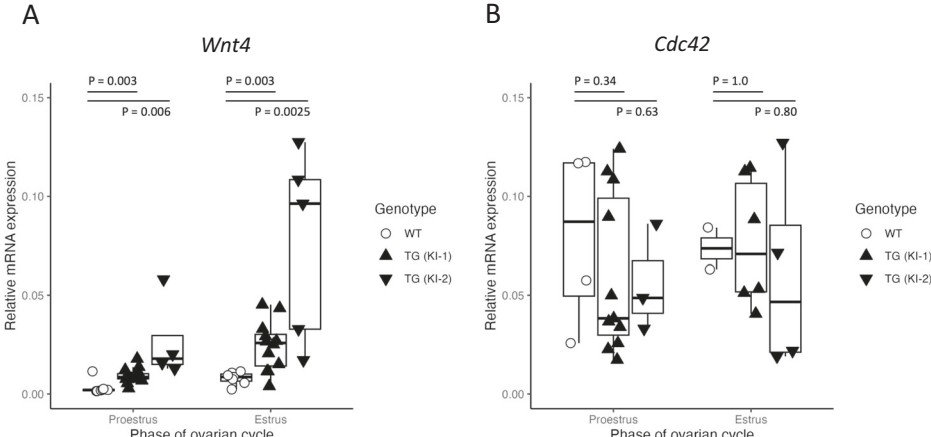

**Fig. 2 | SNP influences expression of *Wnt4* in the uterus. A** Both independently generated knock-in (KI) lines had significantly higher uterine expression levels of *Wnt4* in the proestrus phase (KI-1: $p = 0.003$, KI-2: $p = 0.006$) and estrus phase (KI-1: $p = 0.003$, KI-2: $p = 0.0025$) of the ovarian cycle compared to the wildtype (WT). In all cases, significance was evaluated by Wilcoxon-Mann-Whitney rank sum test. Number of independent samples (i.e. individuals) used for *Wnt4* measurements: proestrus: 7 WT, 15 KI-1, 4 KI-2; estrus: 7 WT, 12 KI-1, 5 KI-2. **B** Expression of other adjacent genes to the SNP such as *Cdc42* did not differ between genotypes ($p > 0.3$). Number of independent samples (i.e. individuals) used for measurements of *Cdc42*: proestrus: 4 WT, 11 KI-1, 3 KI-2; estrus: 2 WT, 6 KI-1, 4 KI-2. Statistical significance was evaluated by the Wilcoxon-Mann-Whitney rank sum test. Boxplots show the median (central line) and interquartile range (50% of the data fall within the box), with the whiskers covering the range of data (Q1 or $Q3 + 1.5IQR$).

## Wnt4 upregulation affects endometrial stromal fibroblasts

To determine the uterine cell type in which *Wnt4* is upregulated in the transgenic line, we performed in situ hybridization using RNAscope[22] on proestrus uteri. We found that *Wnt4* is expressed in luminal and glandular epithelium in both genotypes. In contrast, in the transgenic line *Wnt4* is also strongly expressed in the stromal cells specifically underneath the luminal epithelium (Fig. 3A). To further solidify this finding, we isolated primary endometrial stromal fibroblasts during late proestrus from transgenic and wild type mice and measured the respective expression levels of *Wnt4* by qPCR. Primary cells isolated from the transgenic line exhibited a 2.71 log2 fold upregulation of *Wnt4*, relative to those of a wild type (Fig. 3B; P = 0.0004; see Supplementary Fig. S3 for the purity of stromal cell extractions). Stronger upregulation of *Wnt4* in the primary endometrial stromal cell culture relative to the bulk uterine transcriptome is consistent with the former being enriched for the affected cell type.

## Transcriptional upregulation dominates over downregulation

To understand the overall uterine consequences of the variant, we compared the uterine tissue-transcriptomes between WT and TG mice in proestrus (Supplementary Fig. S4). This approach revealed the systemic modification of the uterine transcriptome, which is likely a consequence of the increased *Wnt4* expression in the individuals carrying the transgenic alleles.

Uterine transcriptomes confirmed upregulated *Wnt4* expression in transgenic individuals (Fig. 4A; adj. *P* < 0.001; 1.8 log2 fold higher expression in the TG strain; Supplementary Data File 2), with no significant difference in *Cdc42*, or other genes in the region of the SNP, congruent with qPCR results. Apart from Wnt4, we detected 142 genes to have significantly changed proestrus uterine expression levels in the TG relative to WT line (Fig. 4A; adj. *P* < 0.05; Supplementary Data File 2). Of these, five times as many genes are upregulated (119) than downregulated (23), a significant enrichment for increased expression (Fig. 4B; *P* < 0.001). In the following, we focus on the major GO groups of genes with changed expression, and their biological roles in the uterus. Their pathogenic potential will be addressed in the discussion.

## Downregulated genes are enriched for proliferation pathways

The significantly downregulated genes showed very clear enrichment in genes related to proliferation, mitosis, and DNA damage repair (e.g.

*Mki67, Racgap1, Tacc3, Ankle1, Rad51, Melk;* Fig. 4B). Specifically, hallmark gene set enrichment[23] was found for processes involving the cell cycle including the G2/M checkpoint, DNA damage repair, and genes associated with mitotic spindle assembly (Supplementary Fig. S5 A; adjusted *p* < 0.001; Supplementary Data File 3). In addition, upstream proliferation effectors were among hallmark gene set enrichments in downregulated genes, including targets of proliferation-inducing E2F and Myc transcription factors, unfolded protein (a cellular ER stress response), and Mtorc1 complex. We also detected a significant downregulation of the progesterone metabolizing enzyme *Akr1c19* (mouse ortholog of human *AKR1C1*) and serine protease inhibitor *Spink12*, known to be expressed in various epithelia[24]. Immunohistochemistry with anti-Mki67 antibody confirmed that the protein expression of Mki67 (proliferation marker) in proestrus is pronounced in the luminal epithelium of wild type, while almost lacking in the luminal epithelium of transgenic uteri. Glandular and stromal compartments show low expression, which does not differ between genotypes (Fig. 5; for quantitative image analysis see Supplementary Fig. S7 and Supplementary Data File 4).

## Upregulated genes are enriched for progesterone response

Upregulated genes show substantially more complex effects of the mutation than the downregulated genes (Fig. 4C). The enriched hallmark gene sets predict upregulation of pathways related to epithelial-mesenchymal transition, myogenesis, hypoxia, coagulation, KRAS-signaling, and TNFA-signaling via NFKB, among others (Supplementary Fig. S5; adj. *P* < 0.001; Supplementary Data File 3). These pathways characterize uterine processes active during the ovarian cycle, with the upregulated genes suggesting onset of decidualization (*Zbtb16, Tcf23, Crebrf*), increased invasibility of uterus by the trophoblast (*Fst, Rgs2, Itgb3, Itga8,* and *Lim2*)), hemostasis/vascularization (*Rgs2, F5, Serpine1, Mfge8*) and the increased glucose utilization (*Irs2, Igfbp2, Pdk4, Hmgcs2, Slc2A12;* Fig. 4C). Specifically, the above uterine processes are associated with increased progesterone signaling, such as occurs in the early secretory phase of human menstrual cycle, but in mice is restricted to early pregnant or pseudopregnant state. As these processes are not well reflected in the existing gene ontology categories (e.g. see decidualization: GO:0046697) we briefly summarize the support for their inference in the following paragraphs.

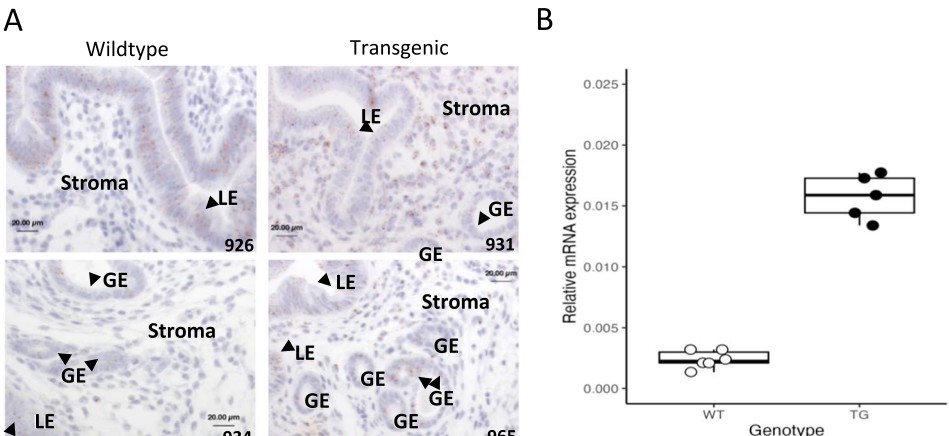

**Fig. 3 | Changes in *Wnt4* proestrus expression are concentrated in the uterine stroma. A** RNAscope localized expression of *Wnt4* in the luminal epithelium of both WT in TG lines during proestrus. Expression was also identified in the stroma for the transgenic line but was not detectable for the wildtype. Luminal epithelia (LE), glandular epithelia (GE), and stroma are indicated. Two biological replicates are shown for each line. Wnt4 is stained a light brown, and the nucleus is a light indigo. Scale bar in each image measures 20 μm. **B** The concentration of *Wnt4* expression differences in the stroma were confirmed with qPCR of primary endometrial stromal fibroblast cells (6 independent samples for WT, 5 for TG; See Fig. S3 for cell purity). Statistical significance was evaluated by the Wilcoxon-Mann-Whitney rank sum test. Boxplots show the median (central line) and interquartile range (50% of the data fall within the box), with the whiskers covering the range of data (Q1 or Q3 + 1.5IQR).

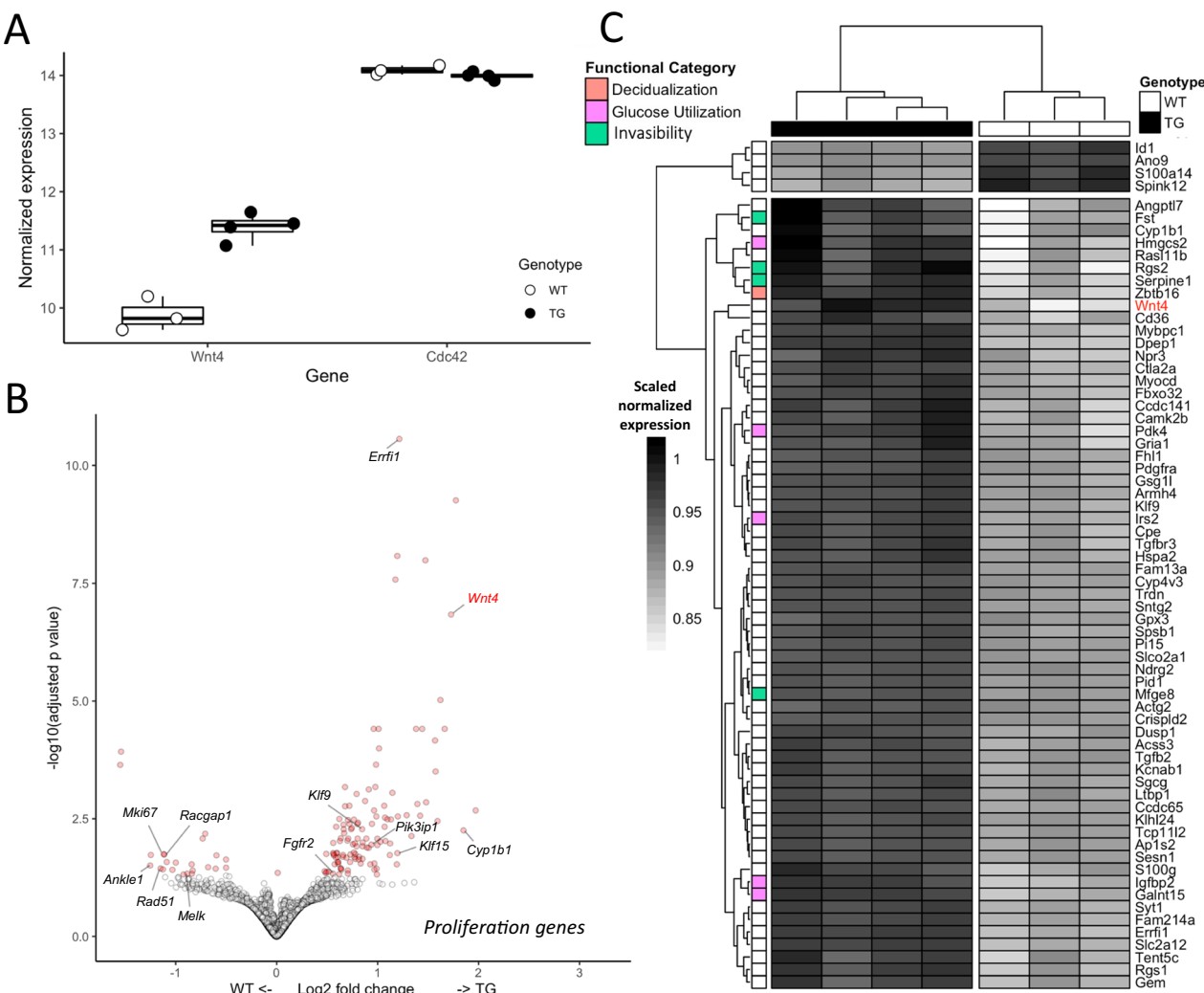

**Fig. 4 | SNP mediated differential expression in the transgenic proestrus uterus. A** Effect of the SNP on flanking genes, with differences in *Wnt4* expression between the wildtype and transgenic lines, but not *Cdc42*, was confirmed in the RNAseq analysis. **B** Volcano plot showing significance and effect size distribution. Log2 fold change values are calculated with respect to TG such that downregulated genes have a negative value and upregulated genes have a positive value. Significantly differentially expressed genes (adj. $P > 0.05$) are in red. Genes associated with proliferation are labeled in black and *Wnt4* is indicated in red. **C** Heatmap of significantly differentially expressed genes with an absolute log2 fold change > 0.5. Genes with functions of relevance to reproductive phenotypes were among those with the greatest increase in expression. All plots are based on independent transcriptomes of 4 TG and 3 WT individuals Boxplots show the median (central line) and interquartile range (50% of the data fall within the box), with the whiskers covering the range of data (Q1 or Q3 + 1.5IQR).

We detect upregulation of progesterone-associated genes, in particular those related to modulation of uterine epithelial proliferation (e.g. *Errfi1, Pikp3*[25], *Fgfr2*[26], *Cyp1b1, Klf9, Klf15*,[27,28] Fig. 4B). The most significantly upregulated among these is *Errfi1*, the Errb feedback inhibitor 1, shown to inhibit the action of epidermal growth factors[29]. In the uterus, *Errfi1* mediates progesterone activity by inhibiting the mitogenic action of Erbb2 (receptor tyrosine kinase) and opposing the estrogen-driven endometrial proliferation in the presence of progesterone[30]. Interestingly, an estrogen metabolizing enzyme *Cyp1b1* is also upregulated in TG, potentially contributing to the local metabolism of mitogenic estrogen. The enhancement of proliferation inhibitors is consistent with the downregulation of proliferation markers such as *Mki67* in epithelium (Fig. 5 and Supplementary Fig. S6).

Apart from the effects on proliferation, a set of upregulated genes is indicative of the onset of decidualization, a progesterone-induced uterine endometrial transformation that in mice does not normally occur in nonpregnancy (but does in humans). Among these genes are *Zbtb16*, a transcriptional repressor of cell cycle, shown to mediate progesterone receptor-driven decidualization of endometrial stromal cells in human;[31] a decidualization-associated transcription factor *Tcf23*[32], and *Crebrf*, a factor involved in regulation of stress response, whose expression is known to increase at implantation sites[33].

Third, a set of upregulated genes implies the enhancement of processes of receptivity and invasibility. These genes include Folistatin (*Fst*), whose uterine-specific knockout in mouse reduces the responsiveness of the luminal epithelium to estrogen and progesterone[34] and integrins (*Itgb3, Itga8*, and associated *Lim2*), implicated repeatedly in embryo attachment[35], wound healing and vascularization[36], the epithelial-to-mesenchymal transition (EMT) and stromal invasibility by cancer[37]. Also upregulated in the transgenic mice are the regulators of G-protein signaling, *Rgs1* and *Rgs2*, previously linked to regulation of vascular tone in implantation[38]. Further indication of early vaso-activation is the upregulation of genes with endothelial or peri-endothelial expression (*Serpine1*[39], *Mfge8*[40]).

Finally, we note an increased expression of genes involved in glucose uptake and utilization, characteristic for decidualization and implantation (*Irs2, Igfbp2, Pdk4, Hmgcs2, Slc2A12;* Fig. 4C)[41,42]. Glucose

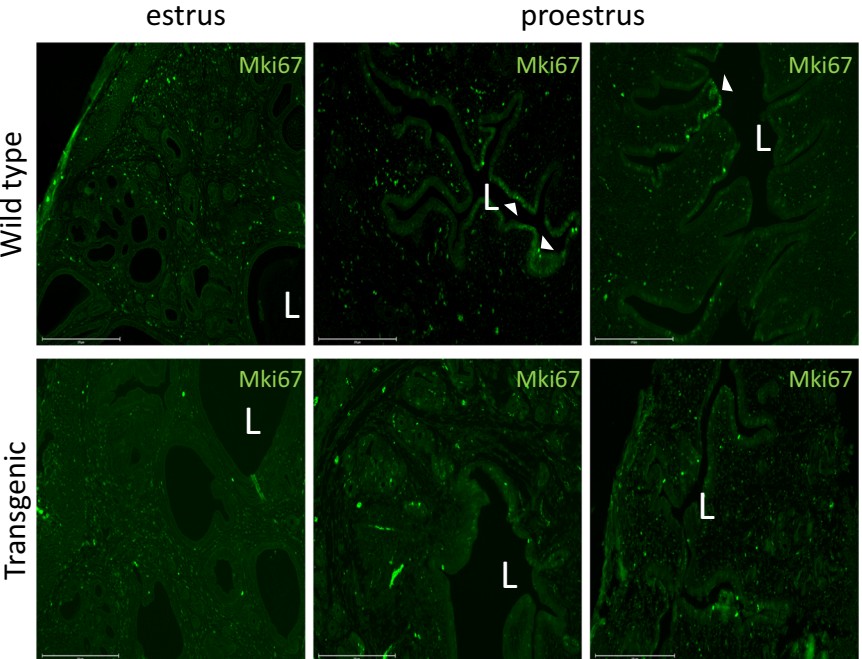

**Fig. 5 | Proliferation marker is expressed primarily in the proestrus luminal epithelium of WT.** Marker of proliferation, MKI67 is expressed in WT in the proestrus luminal epithelium and decreases in estrus. In TG animals, we do not detect epithelial expression in proestrus or estrus. Proliferating luminal epithelium (L) is marked by arrowheads and is most prominent in proestrus of the wild type. Scale bar corresponds to 275 µm. The experiment was repeated 3 times with different individuals and consistent results (not shown).

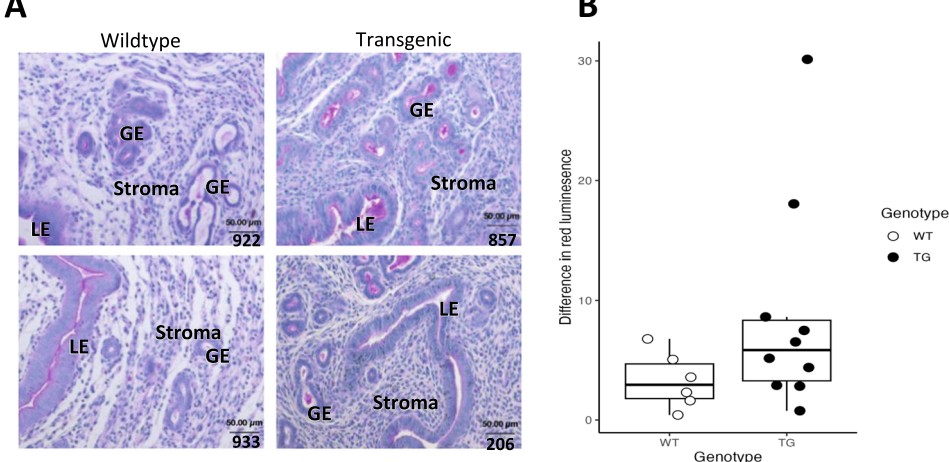

**Fig. 6 | Transgenic line shows higher content of polysaccharides in proestrus uterine tissue. A** Periodic acid - Schiff (PAS) stain of proestrus uterus from wildtype and transgenic lines. Pink color in glandular lumen or cellular cytoplasm indicates polysaccharides. Scale bar in each image measures 50 µm. **B** Quantification of difference in red luminescence between the serial sections treated with α-amylase to digest glycogen and the nontreated PAS-stained sections detects a trend towards a stronger reduction in signal in TG, indicating higher concentration of glycogen ($P = 0.12$; Wilcoxon-Mann-Whitney rank sum test). Number of independent samples (i.e. individuals) used is 4 WT and 10 TG. Boxplots show the median (central line) and interquartile range (50% of the data fall within the box), with the whiskers covering the range of data (Q1 or Q3 + 1.5IQR).

from the maternal blood as well as stored as glycogen enables the metabolically demanding restructuring processes of decidualization, serves as critical nutritional support for developing embryo and enables protein glycosylation of abundant glycoproteins in the uterus, including those of increased glandular secretions. Upregulation of the genes involved in these processes is consistent with histological results, showing a trend towards transgenic uteri containing higher total amount of polysaccharide and increased amounts of glycogen and glandular secretions in proestrus (Fig. 6; $P = 0.12$; see Supplementary Data File 4). In accordance with this observation, we note upregulation of *Galnt15*, an enzyme catalyzing protein glycosylation, in particular, of mucin.

These progesterone-associated changes in gene expression are not accompanied by a significant increase in serum progesterone levels (Supplementary Fig. S7, $P = 1$, Supplementary Data File 4), although there was a greater range in progesterone expression in TG compared to WT. We also did not find increased progesterone receptor gene expression in the uterus during proestrus ($P = 0.36$).

This indicates that the Wnt4 mediates an aspect of progesterone-induced changes in normal uterine biology, as suggested by Franco[43].

## ß-catenin-dependent pathway is not activated

Given the *Wnt4* upregulation and the observed transcriptional signature, we asked whether Wnt signaling pathway used is ß-catenin dependent canonical pathway, involving inhibition of ß-catenin degradation and promoting its translocation into the nucleus, where it acts as a transcription factor for various target genes[44]. Wnt4 is known to signal via canonical as well as several non-canonical pathways in reproductive contexts as shown in the gynecological diseases[21]. Beta-catenin-dependent canonical Wnt4 signaling is crucial for uterine decidualization in the mouse[43]. We used immunohistochemistry in uterine tissue to examine whether the upregulation of *Wnt4* expression coincides with changes in nuclear localization of ß-catenin. We detected no nuclear localization of ß-catenin in the uteri of transgenic or wildtype mice during proestrus or estrus (Supplementary Fig. S8A, B; Supplementary Data File 4). Consistent with this result, there is no upregulation of the enhancer factor *Lef1* or member of the ß-catenin dissociation/degradation complex *Axin2* which are common targets of ß-catenin signaling (Supplementary Fig. S8C and Supplementary Data File 2), concluding that *Wnt4* likely causes described changes via one of the non-canonical pathways. Which exact pathway is involved will require further functional analysis. Some indication that the Ca2+/CamkII pathway may play a role comes from upregulation of *CamkIIb* in the transcriptome (Supplementary Fig. S7D), whereas there is no difference in *Mapk8* expression suggesting the JNK pathway is not involved (Supplementary Data File 2). However, the exact downstream mechanism of increased *Wnt4* activity requires a separate focus in the future.

## The uteri of transgenic mice manifest increased size during estrus

Increased *Wnt4* mRNA expression across proestrus and estrus is correlated with a 34.7% increase in the uterine cross-sectional area in the transgenic animals in estrus phase (Fig. 7; $P = 0.04$, two-tailed t-test). Given the downregulated luminal epithelial proliferation in proestrus, this volume increase is likely attributable to increased glandular activity, glycogen storage or oedema.

## Wnt4 is not upregulated in pregnancy, yet transcription differs

Analysis of *Wnt4* uterine expression in early pregnancy revealed that there is no difference between the two genotypes (7.5 dpc;

Supplementary Fig. S9A; $P = 0.70$). As *Wnt4* expression increases during decidualization in both genotypes, it is unclear from the bulk transcriptome, whether the differential expression diminishes because of a changed *Wnt4* regulation, or whether a specific cell subpopulation is overwhelmed by other *Wnt4* expressing cells as decidualization progresses. To further address the effect of the allele on mouse pregnancy, we asked whether any gene expression changes are maintained in late pregnant uterus. To include the changes potentially relevant for pregnancy, we used the tissue from the maternal-fetal interface at this time point. Comparison of the maternal-fetal interface transcriptomes at 17.5 dpc between transgenic and wildtype revealed no differences in *Wnt4* expression (Supplementary Fig. S9B; Supplementary Data File 2). In general, maternal fetal interface transcriptomes at 17.5 dpc displayed greater similarity to each other than the uterine proestrus samples (Supplementary Fig. S10A, B). Some of the similarity may be explained by the reduced representation of the uterine cells in the transcriptomes. Nevertheless, enriched hallmark gene sets were indicative of similar aspects as found in the proestrus uterine dysregulation in that they predict upregulation of pathways related to epithelial-mesenchymal transition and myogenesis (Supplementary Fig S10C; Supplementary Data File 3), hence we suggest that these differences could be associated with Wnt4 effects in the preimplantation stage.

As human polymorphism is associated with pregnancy length, we examined mouse gestational length and litter size in the transgenic lines. We did not observe differences in gestation length (Supplementary Fig. S11A; $P = 0.33$: Supplementary Data File 4) or litter size (Supplementary Fig. S11B; $P = 0.66$; Supplementary Data File 4) in either transgenic line, relative to wild type.

Overall, the observed proestrus expression profile in transgenic mice suggests that the activation of *Wnt4* by estrogen receptor in alternate allele genotype induces gene expression changes highly similar in spectrum to those mediated by progesterone in early pregnancy in mice and human, or in human secretory phase. However, these changes occur without progesterone increase. In particular, we observe the downregulation of epithelial proliferation and upregulation of genes associated with stromal processes: decidualization, adhesion, extracellular matrix modification, glandular activation, invasibility, and vascularization. While several genes characteristic for decidualization are upregulated, the pattern

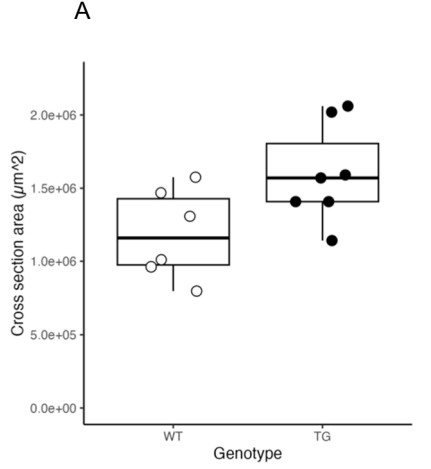

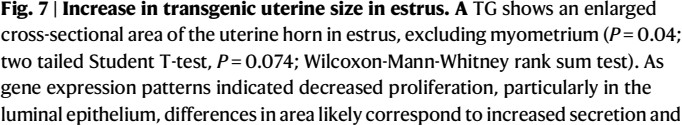

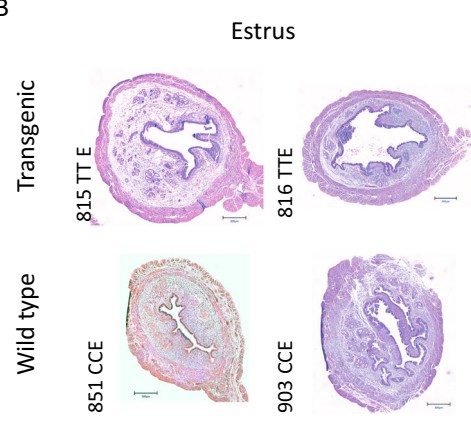

**Fig. 7 | Increase in transgenic uterine size in estrus. A** TG shows an enlarged cross-sectional area of the uterine horn in estrus, excluding myometrium ($P = 0.04$; two tailed Student T-test, $P = 0.074$; Wilcoxon-Mann-Whitney rank sum test). As gene expression patterns indicated decreased proliferation, particularly in the luminal epithelium, differences in area likely correspond to increased secretion and luminal area along with possible oedema. Six independent samples of WT mice and 7 of TG mice were used. Boxplots show the median (central line) and interquartile range (50% of the data fall within the box), with the whiskers covering the range of data (Q1 or Q3 + 1.5IQR). **B** Two H&E-stained cross-sectional uterine samples of each genotype. Scale bar in each image measures 300 μm.

doesn't correspond to full decidualization. For example, the influx of the immune cells (uNK) could not be detected (Supplementary Fig. S12). The observed changes precipitate the overall uterine volume increase in transgenic animals. Despite the lack of significant effect on the total pregnancy length in mice, we do observe continued dysregulation in the uterine transcriptome in the maternal-fetal interface of late pregnancy (17.5 dpc).

## Discussion

In this study, we report a functional analysis of the SNP rs3820282 that has been associated with multiple disorders of female reproduction and reproductive tissues. We have previously shown that the human alternate allele generates a high affinity binding site for the estrogen receptor alpha[13]. The present study aimed to uncover the immediate consequences of this allele substitution. Having created a transgenic mouse line with the human alternative allele, we took advantage of the natural estrogen dynamics during the mouse ovarian cycle to unravel the spatio-temporal context of the variant effects, and their direct consequences on gene expression. We found that the transgenic variant causes upregulation of *Wnt4* in endometrial stromal cells, coinciding with the late proestrus estrogen peak. Importantly, the results demonstrate that the transgenic variant activates uterine *Wnt4* expression following the preovulatory estrogen peak independently of the presence of conceptus. The numerous changes in the downstream uterine gene expression suggest effects on two general processes: the suppression of epithelial proliferation, and the upregulation of pathways involved with embryo receptivity and the uterine susceptibility to invasion.

Observed changes should be viewed in the context of normal cycle-related gene expression. The mammalian uterus undergoes remarkable changes throughout the pregnant as well as nonpregnant cycle, mediated by the fluctuations in the ovarian steroid hormones estrogen and progesterone, and their respective receptors[45,46]. The transition from proestrus to estrus following the estrogen peak is characterized by large uterine transcriptional change, precipitating tissue remodeling and changes in cell adhesion and proliferation. Uterine stroma and epithelium react to altering hormone levels interdependently, as shown for example by the requirement of stromal Esr1 for epithelial estrogen action[47].

Uterine *Wnt4* expression too is estrogen-responsive[48] and correspondingly, there are several Esr1 binding sites in the Wnt4 promoter. The single-nucleotide polymorphism rs3820282 investigated here further modulates estrogen responsiveness of *Wnt4* in uterine stroma, specifically around the preovulatory estrogen peak in late proestrus and into estrus. The pattern of uterine transcriptomic change suggests an early onset of implantation-related uterine processes in transgenic mice, including downregulation of epithelial proliferation, enhanced stromal decidualization, increased glucose utilization and increased potential for invasibility. These uterine changes are normally associated with progesterone increase. Consistent with this apparent switch from estrogen towards progesterone dominance is also the upregulation in transgenic animals of estrogen-metabolizing enzyme *Cyp1b1*, and downregulation of the progesterone-metabolizing enzyme *Akr1c19*. Such upregulation of processes associated with progesterone increase occurs naturally in the human non-pregnant cycle but is only characteristic of pregnancy (or pseudopregnancy) in mice, thus appearing early in the transgenic line, and without systematic progesterone increase.

In uterine decidualization, Wnt4 acts as a downstream effector of progesterone signaling and is thus responsible for some of the effects otherwise associated with progesterone upregulation. Franco and coauthors[43] showed that conditional ablation of uterine Wnt4 significantly reduced embryo implantation and increased apoptosis leading to defect in stromal cell decidualization in pregnant mice. They concluded that Wnt4 promote stromal cell

differentiation survival, mediating progesterone effects. The present results are congruent with these findings, with an exception that the positive modification of *Wnt4* expression is induced by proestrus estrogen peak.

Implantation is a critical event, and in species with spontaneous ovarian cyclicity a set of recurring transformations of the reproductive tract evolved in its preparation (i.e., the uterine cycle). Some of these changes precede the presence of the conceptus and thus occur also in the sterile, nonpregnant cycle (e.g. edema), whereas others require copulation or the presence of conceptus. In humans and many primates, the preparatory changes are extensive and include full stromal decidualization, which in almost all non-primate mammals requires additional triggers, associated with the presence of conceptus. The more widespread mammalian preimplantation uterine changes include actin cytoskeleton remodeling and reorganization of focal adhesions and junctional complexes in stromal cells, changes in apical membrane and loss of epithelial polarity, increase of glycogen production in luminal epithelium and stroma, as well as changes in vascular permeability and angiogenesis[49–51]. Several of these processes are mediated by progesterone, coupling the ovarian and uterine cycles; therefore, the observation of enhanced pro-implantation signature downstream of *Wnt4* expression is plausible. A surprising aspect is that these progesterone-induction-like changes should precede the progesterone-dominated luteal phase in mice, likely even induced by the estrogen. In mice, edema and angiogenesis increase during estrus (coinciding with increase in *Wnt4* expression), while further uterine remodeling occurs in days following copulation and fertilization. The upregulation pattern in a proestrus transgenic mouse therefore suggests an advanced induction of the preimplantation processes, which in absence of the conceptus, is broadly reminiscent of the human condition of spontaneous decidualization. The early expression of *Wnt4* may enhance the receptivity by increasing the *invasion susceptibility, i.e. invasibility* of the uterus for implantation, suggesting a possible explanation for the positive effect of this variant on human pregnancy outcomes[13].

Predicting the comparability between model organisms and humans requires understanding of the shared, and accounting for the species-specific mechanisms, which have diverged during evolution. For example, the mouse does not shed decidual cells in non-pregnant cycle, and also lacks the direct access from uterus to the peritoneum due to the ovarian bursa, therefore the predicted effects on endometriosis must be addressed by cell introduction in the future. While we do not yet have a full mechanistic understanding of differences between mouse and human ovarian cycles, we know that many aspects of uterine receptivity are conserved across mammals[49]. This is due to a shared evolutionary origin of invasive implantation in the stem lineage of placental mammals[52], despite partly independent origins of mechanisms controlling subsequent gestation length[53]. The presented consequences of *Wnt4* upregulation are driven by the dynamics of ovarian preovulatory estrogen peak, which is shared between mouse and human. We also show that at least at the level of gene expression, some of the consequences persist into pregnancy. This speaks for the relevance of the present findings for the human condition. Understanding the downstream effects of a SNP in the mouse uterus may therefore enable prediction of the spatio-temporal contexts in which this SNP exerts its effects on human pregnancy and physiology. These questions will necessarily await future studies on human cells and tissues.

Our findings provide the most direct mechanistic evidence to date for a link between this particular SNP and reproductive traits, even if its ultimate effects on final phenotypes (human pregnancy and disease) remain to be shown. The results imply a two-pronged effect of the variant: suppression of proliferation in the epithelium and enhancement of pro-implantation/invasion processes in the stroma. The association of the SNP with cancer and endometriosis

may involve one or both of these. We suggest that the reported correlated associations of the variant with reproductive cancers and endometriosis may be mediated by the second prong: the enhancement of pro-implantation processes, which increase the permissiveness of stroma to invasion by embryo, but also by cancer and ectopic endometriotic tissue. This suggestion is inspired by the correlation across species between endometrial permissiveness to embryo implantation and vulnerability of peripheral stroma to cancer invasion, where species with more invasive placentation are more prone to invasion by metastases[54–56]. The latter is a known phenomenon at the level of species differences. Here we hypothesize that this relationship is not only a species-level phenomenon but also applies to variation within human populations; individuals with increased uterine embryo invasibility may be more prone to cancer metastasis and endometriosis.

Supporting the increased likelihood of cancer invasion in the transgenic line, is the set of the stromal dysregulated genes which match the expression pattern associated in the literature with stromal support of cancer invasion. The processes involved are: matrix remodeling by proteases and cell-matrix adhesion (increased integrins and periostin), vaso-activation (increased *Fst*, *Rgs*[57], *Serpine1*[39]), energy provisioning (increased *Slc2a12*[58]), and promotion of invasion (decreased *Spink12*[59], *Pdgfra*[58] and Wnt signaling[60]). Thus, differences in tumor microenvironmental expression, not in the cancer cells themselves, may explain the association with cancer invasion[61].

While the dysregulation of *Wnt4* is temporally restricted by the estrogen peak, it is not necessarily spatially restricted to the uterus. It is plausible that the underlying processes are shared with other tissues expressing ESR1 receptors, even if the downstream effector genes may differ. The periodically elevated estrogen in the estrous cycle could thereby affect not only the cancer cells themselves, but also the stroma exposed to and interacting with cancer cells. Such menstrual phase-dependent cancer invasibility has previously been suggested with respect to the propensity towards post-resection metastasis in breast cancer[62]. These effects have been suggested to be due to cycle-phase-dependent fluctuation in angiogenic factors (e.g. VEGF) and factors promoting epithelial-to-mesenchymal transition, downstream of estrogen[63]. The results of the present study suggest that vulnerability to this effect may be genetically enhanced by the alternate allele at the locus rs3820282.

Estrogen responsiveness and *Wnt4* activation are common characteristics of many cancers, even if the mechanisms downstream of Wnt4 are context-specific[64,65]. For example, a recent study has described the co-option of WNT4 under the control of ESR1 in the invasive lobular breast carcinoma, with downstream positive effects on tumor cell proliferation, mTOR signaling and mitochondrial function[11,66] – the effects we do not observe in the transgenic mouse uterus, which may be due to the absence of cancer cells.

It is noteworthy that the epithelial prong of proliferation suppression in our mouse model contradicts the expectation for endometriosis. Endometriosis has been associated with the failure of progesterone receptor to induce genes counteracting estrogen-driven cell proliferation in the secretory phase, i.e. the progesterone resistance[15]. The strongest dysregulation of uterine gene expression in endometriosis was detected in the early secretory phase, whereby a significant *Wnt4* dysregulation is rarely reported (but see Liang et al[67]., for *Wnt4 decrease*). Interestingly, endometriosis-associated gene dysregulation in early secretory phase matches dysregulation in the transgenic mouse endometrium in the present study in terms of affected gene-set, above all the progesterone-regulated genes (e.g. *Errfi1*, *Bcl6*, *Cited2*, *Irs2*, *Tgfb2*, *Gpx3*, *Tk1*). However, these genes are dysregulated in the direction opposite to the one previously reported. This is an intriguing result, as the support for the causal role of the locus rs3820282 in endometriosis is strong[16,17]. Several factors

could explain the mismatch between the effect in mouse proestrus, and its predicted effect on endometriosis: the species context (human vs. mouse), or the differing proportions of stromal and epithelial cells between mouse endometrium and endometriotic lesions. Alternatively, the associations with endometriosis and cancer may primarily stem from the effects of the SNP other than those on epithelial proliferation, such as enhancement of pro-implantation/invasion- a hypothesis we propose. This question will require a specifically focused experimental setting in the future.

Wnt4 has previously been proposed as the plausible causal candidate for endometriosis because of its genomic proximity to rs3820282 and its role in the development of the female genital tract[68], which could precipitate a tendency towards reproductive defects[14]. However, functional genetic follow-up on rs3820282 in the context of endometriosis did not confirm this effect. Luong and colleagues[17] and Powell et al[69]. tested the genes flanking rs3820282 and found that the alternate allele was associated with downregulation of LINC00339 (RNA gene absent from the mouse locus) in whole blood and endometrium, and upregulation of *CDC42* in whole blood. WNT4 was not detected in whole blood samples (Brisbane genetics study[70]) and also no effect of the rs3820282 genotype on *WNT4* expression in endometrium was detected. Chromosome conformation capture (3 C) in Ishikawa endometrial cell line found looping of the regulatory region with the SNP to *CDC42* and LINC00339[69]. There are several plausible sources for the discrepancy between these results and our current findings. First, while authors accounted for sample variation due to different stages of the menstrual cycle, only 16 of 132 samples were collected in the early secretory phase, somewhat close to the stage in which we show the effect. Second, the 3 C approach would not reveal the regulation of WNT4 as the SNP is in the *WNT4* intron itself. Third, the Ishikawa cell line is derived from adenocarcinoma and thus has an uncertain cellular identity. These cells have been reported not to express *WNT4*[71] – rendering this line unsimilar to endometrial cells. Thus, the role of *WNT4* as a mediator of the locus' effect could not be excluded by this study. However, it is also possible that the presence of LINC00339 in humans modifies the effects at the human locus, resulting in dysregulation of CDC42 in humans, but not in mice, where the LINC00339 is absent. This option should be considered in future study.

In conclusion, *w*e suggest that antagonistic pleiotropy associated with *rs3820282* derives from a common mechanistic effect, namely an increased tissue support for invasion, occurring in different contexts. In the context of embryo implantation, the variant enhances a normal process. In the context of reproductive cancers and endometriosis, it may not directly affect the origin or proliferation of cancer cells, but rather decrease the host tissue's ability to resist the invasion, or even provide a supportive context. The idea of an evolutionary trade-off between the mammalian invasive implantation and the vulnerability to cancer metastasis is gaining support in recent research[54]. What may be common to both processes is the shared active role of the stromal microenvironment in regulation of invasion, either by the embryo or by cancer cells. The current study adds to the mechanistic understanding of this phenomenon by pointing out that such permissiveness to cancer may not be constitutive but increases in phases of the ovarian cycle with increased estrogen, modifying not only the uterine, but potentially multiple estrogen responsive tissues. This differs from the idea of pre-metastatic niches in that the permissiveness of tissues to cancer invasion is not induced by the cancer cells themselves but is rather a side-effect of evolved uterine permissiveness and thus occurs, without consequences, also in the absence of cancer (or embryo). This enables a potentially different approach to restricting metastasis of estrogen-responsive cancers - by addressing its spatio-temporally local active support rather than modifying a ubiquitous cancer growth process.

## Methods

### Animal husbandry

Animals were kept under 12:12 light:dark cycle and bred within genotype in house. Offspring were weaned at three weeks and reared separately by sex, housed no more than four animals per cage and fed standard mouse chow diet ad libitum. Animals were euthanized at age 3-7 months by asphyxiation (CCHMC) and cervical dislocation (UoV), consistent with animal care protocols at CCHMC and the UoV, immediately prior to tissue harvest. All procedures were approved by the relevant animal care and protection boards (CCHMC: IACUC2016-0053; UoV: Austrian Ministry for Science and Education, Nr. 2022-0.183.680).

### Generating transgenic mice

The aim of gene editing was to replace the mouse allele at the location in the mouse genome corresponding to rs3820282 in humans, with the human allele. The position rs3820282 and its flanking sequences are 98% conserved between the human and mouse genome (Fig. 1). For gene editing, we selected the sgRNAs that bracket the corresponding site in mice based on the on- and off-target scores from the web tool CRISPOR[72]. The selected sgRNA is constructed in a modified pX458 vector that carries an optimized sgRNA scaffold[73] and a high-fidelity Cas9 (eSpCas9 1.1)−2A-GFP expression cassette[74]. Individual sgRNA editing activity is validated in mouse mK4 cells, using a T7E1 assay (NEB), and compared to the activity of Tet2 sgRNA that has been shown to modify the mouse genome efficiently[75]. Validated sgRNAs are in vitro synthesized using MEGAshorscript T7 kit (Life Technologies) as previously described[76]. Injection was made into the cytoplasm of one-cell-stage embryos of the C57BL/6 genetic background using the method described previously[76]. Injected embryos are immediately transferred into the oviductal ampulla of pseudopregnant CD-1 females. Live born pups are genotyped by PCR and then further confirmed by Sanger sequencing.

### Genotyping

DNA for genotyping was isolated from tail clips and a PCR was run using the following primers: fwd: GCCTCAGAGGAATTGCGAGC and rev: GGATAGCCAACAGTGTAGCTGG. We used a high-fidelity polymerase Regular Phusion (NEB (M0530)) with high annealing temperature, and GC buffer. PCR cycling was as follows:

        98 C hold until load samples-
        Hot Start
        1. 98 °C 2'
        2. 98 °C 15"
        3. 65 °C 20"
        4. 72 °C 30"
        5. 30 cycles steps 2-4
        6. 72 °C 5'
        7. 15 °C until emptied

The products were digested with restriction enzyme Tsp451 (NEB R0583), which cuts if the alternate T allele is present or with Msc1 (NEB R0534), which cuts if the C allele is present.

Only clean bands were purified with QIAquick PCR purification kit (Qiagen) and sent for Sanger sequencing.

### Gestation phenotype

To assess the effect on gestational length, the pregnancies of homozygous females bred to the males of the same genotype were observed. The midnight preceding the morning of plug detection was counted as beginning of pregnancy, and the females were monitored to determine the start of parturition as appearance of the first pup. The litter size was assessed after the labor ceased and checked on the next day.

### Tissue Collection

Animals were bred and tissue collected and harvested at two institutions (University of Vienna, UoV, and Cincinnati Children's Hospital Medical Center, CCHMC). Analyses were repeated with samples from each institution independently, and the results have been consistent.

**Proestrus and estrus.** Adult non-mated female mice (aged 2-8 months) were monitored by daily vaginal swabs to determine the stage of estrous cycle[77]. Animals of each genotype at the locus of interest were harvested in proestrus and estrus, respectively. Uterine horns and ovaries were collected from each individual, one of each was placed in 4% PFA for histological and immunohistochemical assessment, the second was flash-frozen and/or stored in RNAlater for RNA extraction.

**Decidualization and pregnancy.** To assess gene expression during decidualization in vivo, we mated females of known genotype (TG or WT) with males of the same genotype and separated them upon detecting the copulatory plug in the morning examination, counting noon of the detection day as 0.5 day post copulation (dpc). We harvested only the pregnant uteri at 7.5 dpc and 17.5 dpc and preserved the tissue by flash freezing in liquid nitrogen for later RNA extraction and qPCR, respectively RNAseq (17.5 dpc).

**Primary cell isolation and culture.** We isolated primary mouse uterine stromal fibroblast cells using a standard protocol[78], with the difference that uteri were harvested from mice in proestrus, as established by daily vaginal swabs[77]. In brief, uterine horns were harvested and digested with pancreatin and trypsin. The detached epithelial sheets were removed from the solution and the remaining tissue underwent two consecutive rounds of digestion in collagenase (30 min each), washing and filtering of the cell-containing media through 40 μm nylon mesh, spinning and seeding in ESF medium. Mouse ESF culture growth medium consists of equal parts of Dulbecco's Modified Eagle Medium and Nutrient Mixture F12 (DMEM/F12) with phenol red, L-glutamine and 4-(2-Hydroxyethyl) piperazine-1-ethanesulfonic acid, N-(2-Hydroxyethyl)piperazine-N'-(2-ethanesulfonic acid) (HEPES), 10% FBS, 0.5 μg/mL amphotericin B, and 100 μg/mL gentamicin. Cells were grown to confluence in 6-well plates. Before the extraction, the growth medium was removed, and the cells were rinsed with PBS.

### Progesterone measurements

Blood was collected immediately at euthanasia, left at RT for 20 minutes and centrifuged at 1100 x g for 30 minutes to separate the serum. The serum was stored in a new vial at −80 °C until used. Concentration of progesterone in blood serum was measured with a competitive ELISA for mouse progesterone (Crystal Chem). Optical density of each replicate was measured with an absorbance reader (SpectraMax ABS by Molecular Devices) at 450/630 nm wavelength. Progesterone concentration was determined in comparison to standards using (SoftMax Pro 7.1 software).

### Histology

**Sample preparation.** Fresh tissues were fixed in 4% PFA overnight at room temperature. They were then washed with PBS and transferred to gradually increasing concentration of ethanol (30-50-70%). Finally, tissues were then processed and embedded in paraffin blocks. For histological examination, slides were cut at 5 μm thickness and stained by hematoxylin-eosin staining using standard protocols.

**Periodic acid—Schiff (PAS) stain.** Paraffin slides were incubated at 60 °C overnight, deparaffinized in 3 changes of Xylene, and rehydrated

in Ethanol of gradually decreasing concentration (100-95-70%) to DI $H_2O$. To assess the glycogen content, neighboring serial sections were treated with either 1 mg/ml of α-amylase (Sigma-Aldrich), to break down glycogen, or DI $H_2O$, keeping glycogen intact, for 30 min at 37 °C. Next tissue was oxidized in 0.5% periodic acid solution (5 min), rinsed in distilled water, and placed into Schiff reagent (15 min; Sigma-Aldrich). For visualization, untreated slides were additionally counterstained in Mayer's hematoxylin (1 min), washed in tap water (5 min). After subsequent washing (5 min), slides were dehydrated, and cover slipped. For analysis, slides without hematoxylin stain were analyzed with the Celleste software (version 6.0.0). Smart segmentation was used to select the uterine tissue. Manual sectioning was then conducted to retain only the largest endometrial area and mean intensity of red luminescence was measured on this area. The difference in intensity of red luminescence was compared between neighboring slides to determine the amount of glycogen broken down by α-amylase and statistical significance was evaluated with a Wilcoxon-Mann-Whitney rank sum test using R version 4.1.1[79].

**Immunohistochemistry.** After incubating at 60 °C overnight, paraffin slides were deparaffinized by washing in 3 changes of Xylene and rehydrated in gradually decreased Ethanol (100-95-70%) and PBS. Antigen retrieval was conducted using citrate buffer (pH 6.0), followed by $H_2O_2$ to remove endogenous peroxidase activity, and blocking serum to avoid non-specific binding. In all cases, non-conjugated primary antibody was made visible with a secondary antibody with fluorescent fluorophore. The following rabbit-raised anti-mouse primary antibodies were used: Anti-Mki67 (Abcam; ab16667, 5 μg/ml) and Anti-ß-catenin (Abcam; ab16051, 1 μg/ml, not nonclear-specific), Anti-Cd45 (Abcam; ab10558, 1:1000), anti-Cd11b (Abcam; ab133357, 1:4000), anti-Vimentin (Abcam; ab92547, 1:20), anti-Ck7(Abcam; ab181598, 1:100). In all cases we used a goat-raised, anti-rabbit, fluorophore-conjugated secondary antibody (Abcam; ab150077, 2 μg/ml). Luminesce of Mki67 was measured using the image analysis software Celleste (ver. 6.0; Thermo Fisher Scientific), using variable "Intensity Green", and the count of positive cells was done using Fiji[80] (ver.2.14.0/1.54 f).

**RNA scope (in situ).** Commercial RNA scope kit and in situ probe for mouse *Wnt4* RNA was purchased from Advanced Cell Diagnostics, Inc. (ACD). In situ hybridization was performed on paraffin-embedded tissue and developed with chromogenic staining, following the manufacturer's instructions.

**Profiling gene expression**
**RNA isolation.** From frozen uterine horn or ovary approximately 20 mg of tissue was lysed with Precellys Evolution homogenizer at 950 × g (6500 rpm), 2 × 20 s with a stainless-steel bead and 0.04 M ditiotheritol. RNA was isolated using the RNeasy mini kit (Qiagen) according to the manufacturer's instructions. RNA was subsequently stored at −80 °C. From confluent primary cells, RNA was isolated using the mirVANA kit (Thermo Fisher Scientific), according to the manufacturer's instructions. RNA was subsequently stored at −80 °C.

**qPCR.** RNA expression profiles across estrous cycle of genes adjacent to the SNP (Cdc42 and Wnt4) were established by qPCR and compared between the genotypes. RNA was treated with TURBO DNase (Thermo Fisher) to remove any genomic DNA and converted to cDNA with High-Capacity cDNA reverse transcription kit (Thermo Fisher) following standard protocols. qPCR was performed with TaqMan primer probes (Thermo Fisher) with VIC dye for the *GapDH* internal control and multiplexed with FAM for *Wnt4*. Proestrus and Estrus samples were run on an AriaMx qPCR with Brilliant III Ultra-Fast master mix (Agilent). 7.5 dpc and decidual cell lines were analyzed with TaqMan Gene Expression master mix (Thermo Fisher) on a Mastercycler realplex (Eppendorf). Relative gene expression ($2^{-\Delta Ct}$) was calculated with the respective associated qPCR analysis software. Statistical analyses and visualization of qPCR results were done using R, version 4.1.1[79]. Statistical significance was evaluated by the Wilcoxon-Mann-Whitney rank sum test. Only samples with values differing by more than 10 SD were removed.

**Transcriptome analysis (RNA Seq).** RNA from four TG and three WT proestrus uteri was extracted as described above. Library preparation and sequencing was conducted by the CCHMC sequencing facility on on an Illumina NovaSeq 6000 to the depth of 30 million paired end reads of 100 bp length. RNA was also from 17.5 dpc maternal-fetal-interface of three TG and three WT with the same method. Library preparation and sequencing was conducted by Novogene on an Illumina NovaSq 6000 to the depth of 15 million paired end reads of 150 bp length. Reads were aligned to the mouse GRCm38 genome[81] with STAR using settings−outFilterMultimapNmax 1−quantMode GeneCounts[82]. Gene expression was analyzed with DEseq2[83] and an adjusted *p* value > 0.05 was used to identify differentially expressed genes, shrunken log2FC values (with apeglm) were used to account for lowly expressed genes[84] (Supplementary Data File 3). Relationships among samples were investigated with variance stabilizing transformed data and batch effects removed with limma[23] using a principal components analysis as well as hierarchical clustering of Pearson correlation between samples. Gene set enrichment[84] was analyzed with the GSEA function in clusterProfiler[85] using the Hallmark collection from the Molecular Signature Database[23], using ranked shrunken log2FC (Supplementary Data File 4). Gene ontology (GO) overrepresentation was analyzed with the enrichGO function in clusterProfiler[85] on significantly differentially expressed genes (*p* < 0.05). Representative GO categories were determined with ReviGO[86,] and these were primarily used for visualization and interpretation. Data was analyzed with R version 4.1.1[79].

**Reporting summary**
Further information on research design is available in the Nature Portfolio Reporting Summary linked to this article.

## Data availability
All raw transcriptomic data generated in this study have been deposited to the Gene Expression Omnibus repository (NCBI) and are accessible under accession code GSE254989. The phenotypic data as well as processed data generated in this study are provided in the Supplementary Information files.

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

## Acknowledgements

The work was funded by the March of Dimes Ohio Collaborative Project to Louis Muglia (#22-FY14-470) and the Austrian Science Fund to MP (FWF #P33540). We particularly acknowledge the help in histology by Gale Macke of Cincinnati Children's Hospital and the support by Elisabeth Rauscher at the University of Vienna. Open Access funding provided by the University of Vienna.

## Author contributions

MP, GPW, and Louis M developed the idea and planned the research, Lisa M, AMZ, CEM-G, and MP performed the experiments and the analyses, Y-CH helped with generating transgenic lines, NM and GD helped with the genotyping, Lisa M, GD, FK and JM were instrumental in maintaining the colony and collecting the tissue, NZ helped with organizing the samples and validation, DS contributed to localization of the variant effects, GZ crucially contributed to interpretation of genomic signal, MP and CEM-G wrote the draft. All authors have reviewed and discussed results, interpretation, and formulation.

## Competing interests

The authors declare no competing interests.
