## [Peer Review File · Nature Communications]

A common allele increases endometrial Wnt4 expression, with antagonistic implications for pregnancy, reproductive cancers, and endometriosisREVIEWER COMMENTS

Reviewer #1 (Remarks to the Author):

This is an exciting study by Pavličev and colleagues that examines if and how a common human single nucleotide polymorphism rs3820282 can account for antagonistic pleiotropy by using CRISPR/Cas9 to introduce the same nucleotide substitution in the homologous site in the mouse genome. The authors demonstrate upregulation of uterine Wnt4 mRNA expression in late proestrus, which they attribute to the fact that the nucleotide substitution introduces an intronic ESR1 binding site. They characterise the impact on pre-pregnancy gene expression but found no impact on gestation length or litter size. Much of the remainder of the manuscript is taken up by discussing how the differentially expressed genes observed in the uterus during proestrus may explain the association between this SNP and endometriosis and reproductive cancer. Taken together, this is an ambitious and challenging study, and the authors should be congratulations on the experimental data so far. However, the functional characterisation of the mutant mice appears rather limited, and opportunities for more concrete biological evidence of the clinical relevance of this SNP may have been missed. For example:

1. Parturition in mice depends on corpus luteum involution and falling progesterone levels, unlike humans. Perhaps mice carrying the altered allele will undergo spontaneous labour if parturition is maintained with progesterone therapy.
2. Several knockout studies revealed the impact of gene perturbations in pregnancy only if animals are subjected to stress inputs. Perhaps a similar approach is warranted here?
3. The impact of the SNP on uterine function may be cumulative over iterative menstrual cycles. Perhaps this can be modelled in mice by inducing one or more artificial decidual reactions before pregnancy or observing pregnancy over several breeding cycles?
4. There are robust protocols in mice to establish endometriosis – why did the authors not use these to examine the propensity of WT and mutant endometrium to establish ectopic lesions?
5. Can the authors envisage a strategy to test that the SNP promotes the invasiveness of cancer in mutant mice?

The authors should also consider the additional comments:

1. Considering the altered gene expression, the authors should monitor circulating estradiol and progesterone levels. I appreciate that they found no impact on ovarian Wnt 4 transcript levels; this does not exclude subtle endocrine perturbations. To my knowledge, WNT4 is also highly expressed in ovaries and adrenal glands, at least in humans.
2. L193: We observe in the transgenic line the upregulation of many genes previously associated with progesterone increase....without observing upregulation of the progesterone receptor. I do not understand this statement. The progesterone receptor in human endometrium is downregulated upon transition from proliferative to luteal phase, yet progesterone responsiveness increases. What is the evidence that higher PR levels equate to more P4 actions?
3. The authors elegantly describe partial decidual changes in mutant mice that resemble some important aspects of spontaneous decidualization, including oedema, in menstruating primates. Proliferative expansion of uNK cells is a hallmark of this process and precedes the emergence of phenotypic decidual cells. Is this also observed in mutant mice? A potential alternative explanation for the association with cancer may be the impact on local immune cell tolerance rather than intrinsic 'invasibility'.
4. I may have missed it but is there unequivocal evidence that ESR1 binds this intronic SNP in vivo?
5. Given the prevalence of this SNP, can the authors demonstrate the difference in expression of WNT4 in human endometrium or primary endometrial stromal cells?

Reviewer #2 (Remarks to the Author):

Results :

In this paper by Pavlicev and coworkers, the Wnt4 mutation rs3820282 creating a estrogen binding site is introduced in mice. I feel that the results are interesting but that there is a lack of mechanistic explanations to the phenotypic changes observed. Expression is enhanced in pre oestrus, and according to RNA scope due to an ectopic expression in the uterine stroma, and induced transcriptomic and anatomical consequences.

Amongst the questions

1. How did the authors control for off-target effects that might inhibit the gene expression at other genomic positions?
2. Could it be that the absence of statistical increase at later stages is due to an increase in a limited number of cells? The estrogen peak in mice is at the end of the proestrus. Is it the reason envisaged by the authors for the phenotype? Since the mutation is introduced in all the cells how the authors explain a cell-specific change in the Wnt4 response. Did they study chromatin accessibility in luminal and stromal cells separately (single-cell ATAC seq could be an approach).
3. Figure 4 is the transcriptome analysis unsupervised? It would be to carry out a systematic bioinformatics analysis with a Gene Set Analysis approach that would consider all the genes and not only the one selected from the statistical threshold (or is it the data presented in Figure S6. If this is the case, what is the consistency with the gene cascades identified by selecting the genes at a given threshold?
4. Also, in the list of deregulated genes even though Cdc42 is unchanged, there could be an impact on nearby genes. Is this the case? Has the region nearby WNT4 been explored in terms of gene regulation.
5. The supplemental Table should be provided in Excel format. In PDF it is very difficult to read them.
6. The discussion of the up-regulated genes and their function could be better positioned in the discussion, rather than in the results.
7. In figure 5 there should be I think a repeat of the explanation in the legend of the abbreviations, LE, GE, even though they are provided in Figure 3.
8. A, B, C are not correctly referenced in the legend of figure S7. D is not mentioned.
9. About the discrepancy mentioned by the authors at the end of the discussion, they do not evoke the possibility that the absence of the LNC LINC00339 could be crucial, since these RNA genes could have a regulatory impact on a complete genomic region, leading possibly to the deregulation of CDC42 in humans, but not in mice. This may be an additional hypothesis for the organizer role of WNT4 in locus.
10. The up regulation of CMK2B in the uterus is intriguing since this gene is not reported to be highly expressed (if at all) in maternal tissues (uterus) or in the placenta. In humans, it is CAMK2D which is expressed at least in placentae and trophoblasts.
11. If the Beta-catenin is not translocated, the authors suggest a non-canonical action of WNT4. Besides the CAMK genes, there are other parts of the pathway that could be interrogated, especially using inhibitors for instance of JNK or inhibitors of disheveled (DVL) to see at least on cell models such as the stromal fibroblasts (Figure 3A) what is the effect of the inhibitors in the cascade mediated by WNT4. As such I think that the topic has not been deeply explored in the present study.

Reviewer #3 (Remarks to the Author):

Pavličev and colleagues investigate the mechanistic link between a very common genetic polymorphism at the WNT4 locus and associated reproductive phenotypes. This is an important study as it seeks to define the causative variant at a locus robustly associated with human pathologies in numerous GWAS studies, and subsequently define the impact of this SNP on cell/tissue biology. The murine SNP model generated will likely be of broad interest to groups in reproductive health and gynecologic oncology.

This study is not extensive in scope but does address critical points in establishing the model and linking rs3820282 directly to Wnt4 expression and downstream phenotypes. As noted below, one key point regarding Wnt4 in estrus vs proestrus needs to be reexamined, and several qualitative descriptions/data need to be examined in greater depth.

Strengths:

- Study confirms in two separate knock in models that the rs3820282 variant drives Wnt4 expression but not Cdc42, which has been an important contention in the field.
- Gene expression data links the rs3820282 variant to likely oncogenic/cancer-related phenotypes.

Concerns:

- The conclusion in lines 129-133 that Wnt4 is only increased in proestrus should be re-examined. Wnt4 does appear to be increased in estrus (all but one mouse in Figure 2 for $p=0.08$, and significantly in the other model in Figure S1). Certainly estrogen is important but other factors may interact with ER during estrus to further drive Wnt4 expression. This does not minimize the potential role for ER as described in this group's prior work, there may be other transcription factors involved as well. Figure S1 should be moved in to Figure 2. Discussion of estrus v proestrus in the section beginning on line 295 should be revisited.
- Line 260-267: the histopathology of WNT4-variant uteri is extremely lacking. Additional IHC and analysis is merited - does the phenotype represent bonafide hyperplasia OR edema etc? The poor quality ki67 may be hindering the assessment of this phenotype. Overall too speculative / vague, and better definition of phenotype from a pathologist is important. In this section, references to estrus v proestrus should be revisited per above.
- For Figure 3, can any stromal v epithelial marker genes be included to confirm cell type enrichment?
- The Ki67 staining in Figure 5 is of poor quality - very high background - and is difficult to assess given that there is no scoring/quantification. H-score or % positivity assessment by a pathologist would be extremely helpful.
- Figure 6: It is unclear how robust the increase in secretions is from only two images without any quantification.
- Line 288: the results arguably suggest the variant allele activates Wnt4 in response to estrogen, but don't really concretely demonstrate that point.
- The normalization used in the heatmap in Figure 4C is unclear. Also, the black v white WT/TG labels are difficult to differentiate vs the grayscale gradient.
- The sgRNAs used for knockin should be provided; information regarding the repair/editing template also seems to be absent.
- The descriptions of endometriosis in the intro and discussion somewhat lack context given the unclear uterine phenotype information in the results, as described above.
- Lines 191-241: a review for every plausibly interesting gene is verbose and unnecessary, and the results should focus on broader regulatory and signaling implications for the identified geneset.
- Line 243: data shown suggests a lack of canonical/b-catenin dependent signaling, but does not directly support activation of any particular non-canonical signaling.
- Relevant but somewhat vague sections re: WNT4 in reproductive health/cell signaling (e.g. lines 84-89, 247-248) could use/reference a recent review on WNT4 and this SNP in reproductive tissues (PMID: 33963381).
- Lines 353-377 is a bit redundant with 317-337; could be streamlined since the discussion section is a little long.
- The terms mutation v polymorphism v variant are used rather haphazardly. Mutation may be appropriate in the context of the mouse model since it was directly generated as a mutation, but when referring to human data/context, this is a polymorphism or variant in the population.
- Pleiotropy discussion in the intro is relevant but diffuse and distracting, and should be shortened.
- The title is too vague ("affects", "implications") and should be more direct.
- Abstract line 12, "estrogen receptor 1" should be "estrogen receptor alpha".

Responses to reviewers

First, we would like to thank all three reviewers for providing really constructive suggestions. On this basis we have made numerous changes that we think have considerably improved the manuscript. In several places where we are not able to extend this manuscript with new questions (especially on human material or specific animal experimental settings), we do take the suggestions to heart for the future work (as noted also in the text). In the following we provide short answers to the single suggestions and comments.

Reviewer #1:

This is an exciting study by Pavličev and colleagues that examines if and how a common human single nucleotide polymorphism rs3820282 can account for antagonistic pleiotropy by using CRISPR/Cas9 to introduce the same nucleotide substitution in the homologous site in the mouse genome. The authors demonstrate upregulation of uterine Wnt4 mRNA expression in late proestrus, which they attribute to the fact that the nucleotide substitution introduces an intronic ESR1 binding site. They characterize the impact on pre-pregnancy gene expression but found no impact on gestation length or litter size. Much of the remainder of the manuscript is taken up by discussing how the differentially expressed genes observed in the uterus during proestrus may explain the association between this SNP and endometriosis and reproductive cancer. Taken together, this is an ambitious and challenging study, and the authors should be congratulated on the experimental data so far. However, the functional characterisation of the mutant mice appears rather limited, and opportunities for more concrete biological evidence of the clinical relevance of this SNP may have been missed. For example:

1. Parturition in mice depends on corpus luteum involution and falling progesterone levels, unlike humans. Perhaps mice carrying the altered allele will undergo spontaneous labour if parturition is maintained with progesterone therapy.

As the reviewer indicates there are a variety of aspects, including mechanism of progesterone withdrawal at parturition, which differ between mouse and human pregnancy. These differences may contribute to the transgenic allele not showing a significant effect on gestational length in mice, as it had been inferred to in humans.

It is possible that altering environmental or hormonal conditions could reveal a phenotypic difference in gestational length in mice. Progesterone supplementation has been previously shown to increase gestational length in wildtype mice (e.g., Herington et al. 2018, doi.org/10.1210/en.2017-00647). Thus, this treatment may not be the best suited to reveal the effects of the transgenic allele which in humans prolongs gestation.

2. Several knockout studies revealed the impact of gene perturbations in pregnancy only if animals are subjected to stress inputs. Perhaps a similar approach is warranted here?

We agree with the reviewer completely. We are aware that mouse pregnancy may be very robust and that one may reveal a protective effect of the transgenic allele on gestational length if one perturbs this system to reflect a more fragile pregnancy. Although we are not currently able to conduct experimental interventions during pregnancy within our current

animal care protocols, we do plan these in the future. In the meantime, our aim with this study has been to fully investigate this mutation without direct animal experimentation. This way in our future research, the interventions can be as targeted as they can be.

To further investigate the influence of the SNP on differences during pregnancy, we have conducted RNAseq of the maternal-fetal interface from mice 17.5 days post coitus. This allowed us to show the effect of the mutation on pregnancy even when it does not precipitate the change in gestational length of mice. We summarize the results in the manuscript.

3. The impact of the SNP on uterine function may be cumulative over iterative menstrual cycles. Perhaps this can be modeled in mice by inducing one or more artificial decidual reactions before pregnancy or observing pregnancy over several breeding cycles?

The potential influence of parity on pregnancy phenotypes is an interesting aspect. Mice used in these studies were reproductively mature and we did not restrict inclusion of individuals based on the number of past pregnancies or record these data. We will consider multiparity along with pseudo pregnant/pseudo menstrual cycles in future experiments that perturb pregnancy as described in the previous responses.

4. There are robust protocols in mice to establish endometriosis – why did the authors not use these to examine the propensity of WT and mutant endometrium to establish ectopic lesions?

We really appreciate the reviewer's suggestions for future research. We will consider following up on this mouse model in the context of endometriosis phenotypes in our continued research program. For this initial study, our focus has been to establish the putative mechanisms through which this SNP influences Wnt4 and subsequently reproductive phenotypes in a broad manner, and to present it to researchers that are working in these fields, as we think there are many implications of these findings, not all that we will be able to address ourselves.

5. Can the authors envisage a strategy to test that the SNP promotes the invasiveness of cancer in mutant mice?

Most available tests focus on invasiveness of cancer cells themselves, in some specific matrix (matrigel for example). In contrast, the invasibility of tissue or cell aggregation by cancer cells requires a more intricate approach. We are in discussions with collaborators to perform these experiments. Tests of invasibility have been developed (Kshitz et al. 2019, doi.org/10.1038/s41559-019-1046-4), although they have not yet been conducted with mouse ESFs. These tests require not just technology, but also the ability of culturing the transgenic ESF cells for long enough, the CRISPR of the immortalized ESF has been difficult so far, but is in development. Directly testing the influence of the SNP on the invasibility of cancer is certainly an exciting (and necessary!) future direction we hope to be funded to pursue in the next step. These experiments are beyond the scope of the current manuscript.

The authors should also consider the additional comments:

1. Considering the altered gene expression, the authors should monitor circulating estradiol and progesterone levels. I appreciate that they found no impact on ovarian Wnt 4 transcript levels; this does not exclude subtle endocrine perturbations. To my knowledge, WNT4 is also highly expressed in ovaries and adrenal glands, at least in humans.

To address this comment, we have collected plasma samples from proestrus wildtype and transgenic females. Progesterone levels were equally low in both genotypes. We do see broader variance in TG, as is not unexpected in transgenics. Together this further supports that the upregulation of transcription of progesterone-regulated genes is driven by Wnt4 upregulation rather than mediated by increased progesterone. We did not measure estrogen at this time, as its levels are rapidly changing in these stages.

We agree it is very likely that other organs might show the effect on Wnt4 expression. In the discussion we consider the importance of Wnt4 expression in other tissues and the implications for the SNP change in their mode of action *“While the dysregulation of Wnt4 is temporally restricted by the estrogen peak, it is not necessarily spatially restricted to the uterus. It is plausible that the underlying processes are shared with other tissues expressing ESR1 receptor, even if the downstream effector genes may differ.”*

2. L193: We observe in the transgenic line the upregulation of many genes previously associated with progesterone increase...without observing upregulation of the progesterone receptor. I do not understand this statement. The progesterone receptor in human endometrium is downregulated upon transition from proliferative to luteal phase, yet progesterone responsiveness increases. What is the evidence that higher PR levels equate to more P4 actions?

We appreciate the opportunity to clarify the patterns of expression changes in relation to progesterone. Our emphasis is that these patterns of expression differences between genotypes overlap with expression changes found in response to progesterone. We mention the lack of change in the PR to indicate that this increase in pattern does not seem to be mediated by an increase in receptor abundance. We have measured progesterone levels in revision directly to substantiate the point as suggested by the reviewer, rather than make indirect claims.

3. The authors elegantly describe partial decidual changes in mutant mice that resemble some important aspects of spontaneous decidualization, including oedema, in menstruating primates. Proliferative expansion of uNK cells is a hallmark of this process and precedes the emergence of phenotypic decidual cells. Is this also observed in mutant mice? A potential alternative explanation for the association with cancer may be the impact on local immune cell tolerance rather than intrinsic ‘invasibility’.

This is an important point that we did not address originally. In the transcriptome, we see no upregulation of immune cell markers. We have in order to address this comment performed immunohistochemistry using pan- immune cell antibody Cd45, as well as Cd11b (macrophages, neutrophils, NKs) and have not detected differences between the genotypes neither in estrus nor proestrus. We have added this information to the text as well as to the supplemental figures.

4. I may have missed it but is there unequivocal evidence that ESR1 binds this intronic SNP in vivo?

In Zhang et al. 2017 along with reporting the association of the SNP with preterm birth, they conducted both computational and experimental validation of ESR1 binding to this SNP. First the inferred sequence-binding affinity was predicted to substantially increase in the minor genotype. Second, the region of the SNP was shown to have open chromatin and histone markers of transcriptional activity in the cultured endometrial stromal cells. Finally, electrophoretic mobility shift assay used probes of both genotypes to show allele-dependent ESR1 binding - more strongly to the TT genotype (equivalent of our transgenic mice) compared to the CC genotype (equivalent of our wildtype mice). The probes were incubated with or without purified ESF to detect the binding, and the ESR1 as ligand was confirmed using anti ESR1 antibody, generating the supershift. So the binding was not shown in vivo, but the evidence from omic analysis and in vitro experiment is highly consistent. We have added these additional details and put greater emphasis on the evidence for differences in ESR1 binding between the genotypes in the introduction:

Importantly, authors have demonstrated in vitro that the alternate allele at this locus introduces a potent binding site for estrogen receptor alpha (ESR1) , using electrophoretic mobility shift assay (EMSA)¹⁹, in accordance with computational prediction. Moreover, this binding site overlaps with open chromatin in immortalized human endometrial stromal cell line, HESC^{24,19}. The information on downstream consequences of this variant are less robust, although this question has been experimentally addressed previously (e.g.,^{17,25}, see discussion).

5. Given the prevalence of this SNP, can the authors demonstrate the difference in expression of WNT4 in human endometrium or primary endometrial stromal cells?

We are very excited to pursue the suggested research in the future and will certainly be attempting to test the presence of the effect in humans. What we could show is that the ESR1 binding site is accessible in human. The preliminary experiments based on very few samples of (unhealthy) primary ESF cells from the natural human population that were available to us, indicated a large amount of noise. This is to be expected, as with humans we deal with highly heterogeneous genetic background across samples. The degree of control (same protocol, same phase of cell cycle etc) and the sample sizes will have to be high to detect the effect, when there is one, or even to exclude it with certainty. An alternate approach we are also exploring is generating transgenic immortalized human endometrial cell lines, to avoid variation in the background. However, human ESF cells have evaded CrisprCas9 so far, making it difficult to conduct these experiments at this stage.

We truly appreciate the reviewer's thoughtful suggestions. We have added this perspective and an outlook of future experiments to the discussion. These experiments are necessary, but beyond the scope of the current manuscript which focuses on the *in vitro* mechanistic consequences of the SNP in mouse endometrium. We consider our results to be a valuable contribution that furthers our understanding of the SNP, and that expeditiously sharing the current information would benefit the scientific community. We share the reviewer's enthusiasm for elucidating phenotypic effects of the SNP in the setting of human defects and plan to conduct the suggested experiments in the next step. However, as they are not critical

for the current research we do not consider their completion a justifiable reason to delay the dissemination of the current findings.

Reviewer #2 :

Results :

In this paper by Pavlicev and coworkers, the Wnt4 mutation rs3820282 creating a estrogen binding site is introduced in mice. I feel that the results are interesting but that there is a lack of mechanistic explanations to the phenotypic changes observed. Expression is enhanced in pre oestrus, and according to RNA scope due to an ectopic expression in the uterine stroma, and induced transcriptomic and anatomical consequences.

Amongst the questions

1. How did the authors control for off-target effects that might inhibit the gene expression at other genomic positions?

To account for potential off-target effects of Crispr/Cas9, we performed independent Crispr experiments to obtain in total two lines with the same specific mutation in the same genetic background (C57BL6). We used sanger sequencing to ensure that individuals had the exact mutation of interest, and no others, in the region of the SNP. Although we cannot exclude the possibility of off-target effects elsewhere in the genome, it is rather unlikely that both lines would have identical off-target effects. As both transgenic lines have the same phenotype, the SNP of interest, which is present in both independently generated lines, is the most likely responsible for the observed changes in gene expression. We have revised Figure 2 to include the results from both transgenic lines and revised the text in the corresponding results section to highlight the reproducibility of the SNP phenotype.

2. Could it be that the absence of statistical increase at later stages is due to an increase in a limited number of cells? The estrogen peak in mice is at the end of the proestrus. Is it the reason envisaged by the authors for the phenotype? Since the mutation is introduced in all the cells how the authors explain a cell-specific change in the Wnt4 response. Did they study chromatin accessibility in luminal and stromal cells separately (single-cell ATAC seq could be an approach).

As mentioned also by reviewer # 3, the lack of statistical difference in estrus stage was likely due to the larger variance among transgenic individuals (a common phenomenon). Increased sample size in the estrus stage did indeed solidify the expression difference of Wnt4 was also significant in estrous in the KI-1 line. We also observe higher Wnt4 expression in both proestrus and estrus in the other independently derived line (KI-2). Both can now be evaluated in the revised figure 2 and the corresponding results sections.

Without analyzing the data at the single cell level, it is hard to address whether the disappearance of the difference in Wnt4 later in the cycle may be due to the change in cell composition. The RNA scope in situ shows a relatively spread-out population- but it could be that it is overwhelmed later on- as Wnt4 increases towards decidualization. Indeed, the differential expression does coincide with the preovulatory estrogen peak and could be

specific to a cell population. Luminal epithelium already expresses Wnt4 (see RNA scope), so the additional upregulation may not be as effective in these cells. We have added a note on this into the manuscript.

3. Figure 4 is the transcriptome analysis unsupervised? It would be to carry out a systematic bioinformatics analysis with a Gene Set Analysis approach that would consider all the genes and not only the one selected from the statistical threshold (or is it the data presented in Figure S6. If this is the case, what is the consistency with the gene cascades identified by selecting the genes at a given threshold?

The primary functional enrichment analysis included in the manuscript was a gene set enrichment analysis (implemented through clusterProfiler R package). As indicated by the reviewer, this approach includes all genes (i.e., not only those considered differentially expressed at some threshold) and identifies functional enrichments (of hallmark gene sets from the molecular signature database) that are enriched among genes with trends toward increased or decreased expression. As suggested, we also conducted a supervised functional enrichment analysis based on overrepresentation of gene ontology categories among significantly differentially expressed genes (adj. $P < 0.05$). These results showed a concordance with the gene set enrichment data, particularly in categories associated with proliferation higher in WT than TG. These results have been added to both Fig S5 and Table S3.

4. Also, in the list of deregulated genes even though Cdc42 is unchanged, there could be an impact on nearby genes. Is this the case? Has the region nearby WNT4 been explored in terms of gene regulation.

We examined the genes approximately 500 kb up and downstream of Wnt4 in our RNAseq data. In our data set, we identified 11 genes in this region, including Cdc42. None of these genes were significantly differentially expressed. We have noted this additional information to the results.

5. The supplemental Table should be provided in Excel format. In PDF it is very difficult to read them.

This is a constraint of the submission templates. We will attempt in the resubmission to provide also the Excel files.

6. The discussion of the up-regulated genes and their function could be better positioned in the discussion, rather than in the results.

We have thoroughly reworked this part to accommodate the reviewers' comments, in part slimming the results, and also moving parts to the discussion. One point to be made (added in the ms) is that the GO terms capturing the female reproductive processes are poorly characterized (judging from the associated gene lists, a thing that the field should probably reevaluate). This hinders a reliable identification of underlying processes with conventional

pipelines. This is why we offer additional analyses of the gene lists of deregulated genes and spend somewhat more time to support the conclusions about deregulated processes.

7. In figure 5 there should be I think a repeat of the explanation in the legend of the abbreviations, LE, GE, even though they are provided in Figure 3.

We have added this information to the Figure legend.

8. A, B, C are not correctly referenced in the legend of figure S7. D is not mentioned.

We have fixed the Figure S7 legend and ensured that all figure parts are described and match their labels.

9. About the discrepancy mentioned by the authors at the end of the discussion, they do not evoke the possibility that the absence of the LNC LINC00339 could be crucial, since these RNA genes could have a regulatory impact on a complete genomic region, leading possibly to the deregulation of CDC42 in humans, but not in mice. This may be an additional hypothesis for the organizer role of WNT4 in locus.

This is a very interesting idea, thank you. We will integrate it into the discussion. Even if we cannot address this in the current setting because mice do not have the LINC00339, it makes for a very plausible possibility to be tested when moving on to the human cells in the next step.

10. The up regulation of CAMK2B in the uterus is intriguing since this gene is not reported to be highly expressed (if at all) in maternal tissues (uterus) or in the placenta. In humans, it is CAMK2D which is expressed at least in placentae and trophoblasts.

We have at this point decided to not follow this particular issue further, as we believe it will require a separate analysis to understand which particular Wt pathway is activated. The difference to the reviewer's expectation may be because we are looking at the nonpregnant uterus (?)

11. If the Beta-catenin is not translocated, the authors suggest a non-canonical action of WNT4. Besides the CAMK genes, there are other parts of the pathway that could be interrogated, especially using inhibitors for instance of JNK or inhibitors of disheveled (DVL) to see at least on cell models such as the stromal fibroblasts (Figure 3A) what is the effect of the inhibitors in the cascade mediated by WNT4. As such I think that the topic has not been deeply explored in the present study.

Indeed, we did not interrogate the Wnt4 pathway deeply as we think that this would require a separate study- given the many different pathways that Wnt4 seems to be involved in across tissues. Our intention was to limit the range by interrogating one major aspect, the beta-catenin-dependence. Much more work will be required to understand the pathway by which WNT4 signals in the uterus in the future. We did additionally look at expression of JNK and found no difference in JNK (Mapk8) expression. We also saw no difference in expression of inhibitors such as disheveled (DVL 1, 2, or 3), APC, GSK, Axin, or Cskn. expression between the WT and TG lines.

Reviewer #3 :

Pavličev and colleagues investigate the mechanistic link between a very common genetic polymorphism at the WNT4 locus and associated reproductive phenotypes. This is an important study as it seeks to define the causative variant at a locus robustly associated with human pathologies in numerous GWAS studies, and subsequently define the impact of this SNP on cell/tissue biology. The murine SNP model generated will likely be of broad interest to groups in reproductive health and gynecologic oncology.

This study is not extensive in scope but does address critical points in establishing the model and linking rs3820282 directly to Wnt4 expression and downstream phenotypes. As noted below, one key point regarding Wnt4 in estrus vs proestrus needs to be reexamined, and several qualitative descriptions/data need to be examined in greater depth.

Strengths:

- Study confirms in two separate knock in models that the rs3820282 variant drives Wnt4 expression but not Cdc42, which has been an important contention in the field.
- Gene expression data links the rs3820282 variant to likely oncogenic/cancer-related phenotypes.

Concerns:

- The conclusion in lines 129-133 that Wnt4 is only increased in proestrus should be re-examined. Wnt4 does appear to be increased in estrus (all but one mouse in Figure 2 for $p=0.08$, and significantly in the other model in Figure S1). Certainly estrogen is important but other factors may interact with ER during estrus to further drive Wnt4 expression. This does not minimize the potential role for ER as described in this group's prior work, there may be other transcription factors involved as well. Figure S1 should be moved to Figure 2. Discussion of estrus v proestrus in the section beginning on line 295 should be revisited.

We appreciate this suggestion. To better evaluate the effect of Wnt4 in the estrus stage we increased the sample size and found that there was statistical support for a significant increase in Wnt4 expression in this stage as well. We have revised figure 2 to depict the data from both lines and the corresponding result section, and integrated this result into the discussion.

- Line 260-267: the histopathology of WNT4-variant uteri is extremely lacking. Additional IHC and analysis is merited - does the phenotype represent bonafide hyperplasia OR edema etc? The poor quality ki67 may be hindering the assessment of this phenotype. Overall too speculative / vague, and better definition of phenotype from a pathologist is important. In this section, references to estrus v proestrus should be revisited per above.

We have repeated the KI67 fluorescence immunohistochemistry and imaging, and also analyzed the images quantitatively, both in estrous and proestrus. The updated images are in the main manuscript and we provide the analyses in the supplementary material. The results support what we see also in the transcriptome, namely the decrease of luminal proliferation

in proestrus in transgenic mice. We do not see signs of apparent edema or hyperplasia, but do see increased glycogen accumulation as explained in the manuscript.

- For Figure 3, can any stromal v epithelial marker genes be included to confirm cell type enrichment?

Luminal epithelial cells can be identified as the first layer of cells adjacent to the lumen, and are recognizable by their distinct morphology, As the morphology of uterine stromal and epithelial cells in the mouse has been so thoroughly established, we chose (also in the face of otherwise extensive revision), to not additionally mark the epithelial cells for this figure. However, we show observe upregulation of wnt4 in cell culture of explicitly separated endometrial stromal cells from transgenic animals, and provide the images in Supplementary material, showing the cell identity as marked by Vimentin (fibroblast marker), with CK7 (epithelial marker) absent, in Supplementary Figures.

- The Ki67 staining in Figure 5 is of poor quality - very high background - and is difficult to assess given that there is no scoring/quantification. H-score or % positivity assessment by a pathologist would be extremely helpful.

Ki67 staining and analysis has been improved and is consistent with the transcriptomic results. Please refer to the answers to the previous questions.

- Figure 6: It is unclear how robust the increase in secretions is from only two images without any quantification.

We have expanded the sample size here to 6WT /10TG, and used image quantification methods to quantify the extensiveness of staining from the pictures and compare it between the two lines of interest. In addition, we used amylase to be able to identify the glycogen content among carbohydrates. Results are presented in the figure 6 and also in Table S4

- Line 288: the results arguably suggest the variant allele activates Wnt4 in response to estrogen, but don't really concretely demonstrate that point.

It is true that we have not experimentally demonstrated that Wnt4 increase is in response to estrogen, apart from the previous experimental evidence cited, and the temporal correlation. We have adapted the parts of the manuscript to reflect this. Please also see the answer to reviewer 1.

- The normalization used in the heatmap in Figure 4C is unclear. Also, the black v white WT/TG labels are difficult to differentiate vs the grayscale gradient.

We have clarified this in the figure legend and in the methods. We have also made the TG label red so that it is easier to differentiate from the grayscale gradient.

- The sgRNAs used for knockin should be provided; information regarding the repair/editing template also seems to be absent.

We have added a figure into the Supplementary material which provides the detailed information on the Crispr/Cas9 experiment.

- The descriptions of endometriosis in the intro and discussion somewhat lack context given the unclear uterine phenotype information in the results, as described above.

We have somewhat reduced the emphasis on endometriosis and importantly, have acknowledged that there is a species difference between mouse and human due to which we cannot, without manipulation, observe endometriosis in mice (lack of menstrual shedding, lack of access by the cells into the peritoneal cavity). Our focus at this stage is at the more immediate mechanisms of the SNP at the level of tissue change. However since this locus has been so strongly associated with the endometriosis, it must be discussed (and in the future, tested).

- Lines 191-241: a review for every plausibly interesting gene is verbose and unnecessary, and the results should focus on broader regulatory and signaling implications for the identified geneset.

We appreciate this suggestion, and in conjunction with similar comments from Reviewer 2, have thoroughly revised this part, reducing the review of genes and moving it, in part, to the discussion.

- Line 243: data shown suggests a lack of canonical/b-catenin dependent signaling, but does not directly support activation of any particular non-canonical signaling.

True. We have adjusted the wording to make this clearer.

- Relevant but somewhat vague sections re: WNT4 in reproductive health/cell signaling (e.g. lines 84-89, 247-248) could use/reference a recent review on WNT4 and this SNP in reproductive tissues (PMID: 33963381).

We have added the references and compacted the section.

- Lines 353-377 is a bit redundant with 317-337; could be streamlined since the discussion section is a little long.

We have streamlined the two sections and made sure they are not redundant.

- The terms mutation v polymorphism v variant are used rather haphazardly. Mutation may be appropriate in the context of the mouse model since it was directly generated as a mutation, but when referring to human data/context, this is a polymorphism or variant in the population.

We have changed our use of mutation to variant when referring to human examples and transgenic variant in the context of our mouse model.

- Pleiotropy discussion in the intro is relevant but diffuse and distracting, and should be shortened.

We have shortened the first paragraph which contained the majority of the discussion on pleiotropy, removing information that was not directly relevant to disease pleiotropy and its implications for the understanding of disease mechanisms.

- The title is too vague ("affects", "implications") and should be more direct.

We have changed the title. However, we did maintain the "implications" as it is at this point an honest description of the situation, given the state of evidence (solid effects at the tissue level, but not yet at the level of its consequences for disease), collected so far.

- Abstract line 12, "estrogen receptor 1" should be "estrogen receptor alpha".

Changed

REVIEWERS' COMMENTS

Reviewer #1 (Remarks to the Author):

The authors addressed my concerns reasonably and constructively. The manuscript is more balanced, and the limitations are clearly articulated. It is somewhat regrettable that the authors were - as stated - not in a position to carry out additional animal experiments.

Reviewer #2 (Remarks to the Author):

I am satisfied with the authors answer.

Reviewer #3 (Remarks to the Author):

This revised manuscript by Pavlicev et al is greatly improved, and the authors' responsiveness to reviewer concerns has improved clarity and robustness of the data presented and associated conclusions. The quantitative analyses of histopathology are an important improvement over the initial submission, but a few minor concerns remain in this context.

- Standard H&E should be included for readers to appreciate variant-associated tissue architecture. Currently, this has to be read in RNAscope, Ki67 and other IF, and PAS staining. This minor addition is important given that the authors do report increased uterine size in the TG mice without a clear driver of the size.
- Ki67 IF quality is greatly improved, yet the analysis using signal intensity is unclear and unconventional. Most analyses use % Ki67-positive to assess fraction of proliferative cells, which would be relevant here to compare luminal epithelium, glandular epithelium, and stroma independently. This would eliminate the need to use glandular epithelium as a background, which is not well supported. QuPath would be a relevant tool to rapidly re-assess their data: <https://qupath.github.io/>, aside from % positive assessment per tissue compartment by a pathologist.
- Related to the above, the Ki67 analysis as provided ignores the stroma, which is where increased Wnt4 expression was most evident by RNAscope.

REPLIES TO REVIEWERS' COMMENTS

Reviewer #1:

The authors addressed my concerns reasonably and constructively. The manuscript is more balanced, and the limitations are clearly articulated. It is somewhat regrettable that the authors were - as stated - not in a position to carry out additional animal experiments.

Reviewer #2:

I am satisfied with the authors answer.

We are very glad that both reviewers found our responses fulfilling their constructive requests and thank for their suggestions in the first round of review.

Reviewer #3 (Remarks to the Author):

This revised manuscript by Pavlicev et al is greatly improved, and the authors' responsiveness to reviewer concerns has improved clarity and robustness of the data presented and associated conclusions. The quantitative analyses of histopathology are an important improvement over the initial submission, but a few minor concerns remain in this context.

- Standard H&E should be included for readers to appreciate variant-associated tissue architecture. Currently, this has to be read in RNAscope, Ki67 and other IF, and PAS staining. This minor addition is important given that the authors do report increased uterine size in the TG mice without a clear driver of the size.

We certainly agree with this comment and have added the H&E-stained histological images for proestrus and estrus stage of both genotypes in to the Figure 7.

- Ki67 IF quality is greatly improved, yet the analysis using signal intensity is unclear and unconventional. Most analyses use % Ki67-positive to assess fraction of proliferative cells, which would be relevant here to compare luminal epithelium, glandular epithelium, and stroma independently. This would eliminate the need to use glandular epithelium as a background, which is not well supported. QuPath would be a relevant tool to rapidly re-assess their data: <https://qupath.github.io/>, aside from % positive assessment per tissue compartment by a pathologist.

- Related to the above, the Ki67 analysis as provided ignores the stroma, which is where increased Wnt4 expression was most evident by RNAscope.

Responding to this point of assessing both sizes and counts of proliferation marker positive cells, we made a reassessment, have increased the dataset and remeasured all sizes with a more advanced software. (the results do not change substantially, but are more solid) Specifically, we added to the present assessment the counts (in percentage) of KI67-positive cells in luminal and glandular epithelia, as well as the stroma. We thank for this comment and the helpful hint towards the tool.